# Phytochemical Profile and Antimicrobial Potential of Propolis Samples from Kazakhstan

**DOI:** 10.3390/molecules28072984

**Published:** 2023-03-27

**Authors:** Jarosław Widelski, Piotr Okińczyc, Katarzyna Suśniak, Anna Malm, Emil Paluch, Asanali Sakipov, Gulsim Zhumashova, Galiya Ibadullayeva, Zuriyadda Sakipova, Izabela Korona-Glowniak

**Affiliations:** 1Department of Pharmacognosy with Medicinal Plants Garden, Lublin Medical University, 20-093 Lublin, Poland; 2Department of Pharmacognosy and Herbal Medicines, Wrocław Medical University, 50-556 Wrocław, Poland; 3Department of Pharmaceutical Microbiology, Medical University of Lublin, 20-093 Lublin, Poland; 4Department of Microbiology, Faculty of Medicine, Wroclaw Medical University, Tytusa Chałubińskiego 4, 50-376 Wrocław, Poland; 5School of Pharmacy, Asfendiyarov Kazakh National Medical University, Almaty 050000, Kazakhstan; 6Department of Pharmaceutical and Toxicological Chemistry, Pharmacognosy and Botany, Asfendiyarov Kazakh National Medical University, Almaty 050000, Kazakhstan; 7Department of Pharmaceutical Technology, Asfendiyarov Kazakh National Medical University, Almaty 050000, Kazakhstan

**Keywords:** propolis, Kazakhstan, hydroethanolic extracts, dendrogram, black poplar, flavonoids antibacterial, *Helicobacter pylori*, urease

## Abstract

In the current paper, we present the results of Kazakh propolis investigations. Due to limited data about propolis from this country, research was focused mainly on phytochemical analysis and evaluation of propolis antimicrobial activity. uHPLC-DAD (ultra-high-pressure-liquid chromatography coupled with diode array detection, UV/VIS) and uHPLC-MS/MS (ultra-high-pressure-liquid chromatography coupled with tandem mass spectrometry) were used to phytochemical characteristics while antimicrobial activity was evaluated in the serial dilution method (MIC, minimal inhibitory concentration, and MBC/MFC, minimal bactericidal/fungicidal concentration measurements). In the study, Kazakh propolis exhibited a strong presence of markers characteristic of poplar-type propolis—flavonoid aglycones (pinocembrin, galangin, pinobanksin and pinobanskin-3-O-acetate) and hydroxycinnamic acid monoesters (mainly caffeic acid phenethyl ester and different isomers of caffeic acid prenyl ester). The second plant precursor of Kazakh propolis was aspen–poplar with 2-acetyl-1,3-di-*p*-coumaroyl glycerol as the main marker. Regarding antimicrobial activity, Kazakh propolis revealed stronger activity against reference Gram-positive strains (MIC from 31.3 to above 4000 mg/L) and yeasts (MIC from 62.5 to 1000 mg/L) than against reference Gram-negative strains (MIC ≥ 4000 mg/L). Moreover, Kazakh propolis showed good anti-*Helicobacter pylori* activity (MIC and MBC were from 31.3 to 62.5 mg/L). All propolis samples were also tested for *H. pylori* urease inhibitory activity (IC_50_, half-maximal inhibitory concentration, ranged from 440.73 to 11,177.24 µg/mL). In summary Kazakh propolis are potent antimicrobial agents and may be considered as a medicament in the future.

## 1. Introduction

Propolis, also called “bee glue” is a natural product of different bee species. Its viscous form is due to the fact that exudates collected from the buds and flowers of different plant species growing in the vicinity of the bee hive are used to produce propolis [1]. These exudates of botanical origin are chewed by the honeybees and mixed later with pollen and the bee’s saliva (containing several enzymes, among them β-glucosidase) and, finally, after the addition of bee wax the raw propolis is formed [1]. Propolis is used as building and sealing material for protecting the hive entrance of the hive and plugging the holes in hive construction as well as covering intruders (insects and small rodents) who died inside the hive in order to avoid their decomposition [2,3].

The characteristic of physiochemical properties (density, color, and odor), as well as the chemical profile and bioactivity of particular propolis samples, is dependent on numerous factors such as plant source (precursor), climate and weather conditions in the year of harvest, and sometimes the harvest time [3,4].

The different main types of propolis are mentioned in the literature [5,6], poplar type, where plant precursor belongs to some *Populus* spp. (e.g., *Populus nigra* or *Populus* species about similar exudates composition such as *P. balsamifera*) and aspen type (*P. tremula* and similar *Populus* species). Both types are characteristic for central Europe, non-tropic regions of Asia, New Zealand and North America; birch type (exudates are collected from *Betula verrucosa* and *B. pendula*, which occurs in Russia and the northern part of Europe); green type (characteristic for Brazil, where the main plant source of propolis is genus *Baccharis*, e.g., *B. dracunculifolia*); red type (characteristic for Brazil, Mexico, Cuba and Nepal, where plants from *Dalbergia* spp. occur, e.g., *Dalbergia ecastaphyllum* or *D. sisoo*); Pacific type (plant precursor of propolis is *Macaranga tanarius* from Indonesia, Taiwan, and Okinawa Prefecture of Japan); Mediterranean type (mainly Plants from *Cupresaceae* family occurring in Greece, Sicily, Malta and other islands). Apart from ‘pure’ types of propolis, there are also observed mixed types of propolis, e.g., aspen–poplar or aspen–birch–poplar.

Propolis has been used in traditional medicine since ancient times. There is evidence that it was used for medicinal purposes in 3000 BC in Egypt [7]. Nowadays, propolis is widely and often used in cosmetology, the food industry, beverages and nutritional supplements, as well as an ingredient in functional food [7]. Herbalists have recommended propolis according to its antibacterial, antiviral and anti-inflammatory activity, which can be used in the treatment of different types of infections as well as duodenal and gastric ulcers [1,8]. Many biological activities of propolis have been confirmed by modern studies, including anticancer [9], antioxidant [10], antileishmanial [11], wound healing [12], anti-inflammatory [13] and immunomodulatory [14,15].

Propolis in different preparations is used worldwide as a potent antimicrobial agent and is active against numerous bacterial strains, but is especially effective against Gram-positive bacteria, and less effective against Gram-negative ones, with *Helicobacter pylorian* as exception [16]. Most of the bioactivities of propolis are related to the presence of polyphenolic compounds in the chemical composition of this natural mixture. It is noteworthy that polyphenols present in propolis, e.g., chrysin, will not only exhibit an antibacterial effect against microorganisms, e.g., *H. pylori*, but also potentiate the activity of antibiotics [17].

Kazakhstan, as a country with a large area and high biodiversity, is an excellent place to collect various natural products. Surprisingly, almost no research has been undertaken so far on propolis from Kazakhstan—in the same Scopus database we found only our previous research [18,19]. In this investigation, one sample of Kazakhstan propolis was analyzed among the rest Eurasian propolis extracts. Therefore, in the research Kazakhstan propolis exhibited strong antibacterial and further research was justified.

For this purpose, we performed (ultra-high-pressure-liquid chromatography coupled with diode array detection, UV/VIS) and uHPLC-MS/MS (ultra-high-pressure-liquid chromatography coupled with tandem mass spectrometry) profiling of 70% ethanol in water extracts (70EEP, ethanol: water, 70:30, *V*/*V*) of ten propolis samples from Kazakhstan as well as evaluated their antimicrobial potential-evaluation of MIC (minimal inhibitory concentration) and MBC/MFC (minimal bactericidal/fungicidal concentration measurements) in serial dilution technique. According to our previous results [18] and expected strong activity against *H. pylori*, we also investigated the potential of extracts as urease inhibitors (one of the most important virulence factors of *H. pylori*). These measurements were based on the evaluation of IC_50_ (half-maximal inhibitory concentration). It is worth adding that the urease inhibition potential of Kazakh propolis extracts was not tested before.

## 2. Results and Discussion

### 2.1. Chemical Composition of Propolis Extracts from Kazakhstan

Results of uHPLC analyses were presented in Table 1 (component identification), Table 2 (relatively presence of components in mass chromatograms) and Table 3 (relatively presence of main components in UV chromatograms). Representative UV chromatograms (280 nm) were also presented in Figure 1.

Components were identified by comparison with previous research [19,20]. Most of the substances belonged to one of four main groups: free cinnamic and hydroxycinnamic acids, hydroxycinnamic acids monoesters, hydroxycinnamic acids glycerides, flavonoids and others. Among all groups of components, in most samples, the highest peaks belonged to hydroxycinnamic acids monoester and flavonoids.

Generally, most of the components were presented in uHPLC-MS/MS chromatograms in negative ionization mode (Table 2). uHPLC-MS/MS is a very sensitive technique, which allows to track a lot of substances. In the same propolis extracts, we observed above 200 singular peaks. However, most of them remained as unidentified traces. For this reason, in the current paper, we presented 155 main components detected in uHPLC-MS/MS chromatograms (Table 1 and Table 2).

Despite the high sensitivity of uHPLC-MS/MS, there were some specific substances, which did not produce ions in negative mode or produce trace amounts of ions in experimental conditions. These components included some known substances (benzoic acid, caffeic acid ethyl ester, ferulic acid benzyl ester, tectochrysin and pinostrobin) and 14 unidentified compounds. Among these components, benzoic acid, caffeic acid ethyl ester and ferulic acid benzyl ester produced ions in negative mode, but, in our experience, their production strongly depends on experimental conditions. Usually, too low concentration and close proximity of easy ionized component on the chromatogram suppress ion production by these components. For this reason, their presence was exhibited only in uHPLC-DAD chromatograms (Table 3) and they were identified by comparison of UV spectra with previous research [19,20,21]. Therefore, most of the components were detected in the uHPLC-MS/MS analysis (Table 1 and Table 2), Table 3 was limited to major substances which were further used in statistical analysis.

The main components of the hydroxycinnamic acids monoesters group included caffeic acid 2-methyl-2-butenyl ester, caffeic acid 3-methyl-2-butenyl ester, caffeic acid 3-methyl-3-butenyl ester and p-methoxy cinnamic acid cinnamyl ester. These components were presented in all samples. Apart from them, high peaks were also observed for *p*-coumaric acid benzyl ester but not for all samples.

In the flavonoid group, most of the components were flavonoid aglycones or their ester derivatives (mainly pinobanksin). The most abundant component was pinobanksin-3-O-acetate. The rest of the common flavonoid aglycones included pinobanksin, chrysin, pinocembrin and galangin.

Among free cinnamic and hydroxycinnamic acids, the most often observed in uHPLC-MS/MS chromatograms were caffeic acid, *p*-coumaric and ferulic acid. The same caffeic acid was observed in all the samples, while the rest of the components were not. The last group included mainly unidentified components and some known, such as benzoic acid. This group of components was significant only in Alamaty-1. The chemical composition of analyzed propolis types as well as plant distribution maps [22,23] and our previous research [18,19] suggested poplar trees as the main plant precursors of Kazakh propolis. Main poplar markers included flavonoid aglycones (pinobanksin, pinbankisn-3-O-actetate, galangin, chrysin, pinocembrin) and methylbutenyl esters of caffeic acid (2-methyl-2-butenyl ester, 3-methyl-2-butenyl and 3-methyl-3-butenyl).

Apart from poplar, the presence of 2-acetyl-1,3-di-*p*-coumaroyl glycerol suggested aspen origin of some samples. This substance is specific marker of *P. tremula* (aspen) as well as some other *Populus* species (e.g., *P. lasiocarpa* [24]). Aspen is widely spread across whole Eurasia [23] while *P. lasiocarpa* is naturally present in China. Outside China, it is rather planted in botanical gardens and parks than easily spreading in the natural environment. For these reasons, presence of 2-acetyl-1,3-di-*p*-coumaroyl glycerol in propolis rather proves the presence of *P. tremula* exudates more than another species.

Previous research exhibited mixed aspen–poplar origin of Kazakh propolis sample [19]. In the current research, most samples exhibited a strong presence of poplars markers (Almaty 1, 4, 6, Bozovoe and Kogaly and Kegen) and lower (Almaty 2, 3, 5) or a strong of aspen ones (Almaty 7). However, there were some unidentified components, which may not be connected with *Populus* genus origin, especially in Almaty-1 (see Table 2 and Table 3). Moreover, a high occurrence of cinnamic acid is also not usual for black poplar trees, but hydroxycinnamic acids are presented in its place [25,26]. According to the plant distribution map [22], *P. nigra* should be rather present in northern Kazakhstan. Therefore, the sample from Bozovoe may exhibit black poplar origin. Propolis from the southern part (Almaty, Kegen and Kogaly) should be rather originated from another balsamic *Populus* tree (e.g., *P*. *laurifolia* or *P. euphratica*). However, literature studies exhibited that the situation of *Populus* genus distribution in Kazakhstan is complex. During the Soviet Union time in the central Asia region, there were introduced many *Populus* species such as *P. nigra*, *P. bolleana* and *P. deltoides* and other cultivars [27]. Moreover, many *Populus* cultivars are known for the easy creation of crossbreed species. In a result in the same Almaty, many crossbreed species were observed (e.g., *P. nigra* × *P. maximowiczii*, *P. nigra* × *deltoides* or *P. laurifolia* × *P. canadensis*) [28]. Dependences between *Populus* bud exudate compositions and their genetic origin were not well investigated and, for this reason, accurate tracking of plant precursors of Kazakhstani poplar type propolis may be very difficult.

Apart from the *Populus* genus, it is worth to add that Kazakhstani propolis may have some other plant sources. In the previous research [19], we observed also the presence of flavonoid ermanin which may be connected with birch origin. However, in the present research, ermanin was only a trace component and the same *Betula* genus may be a marginal plant precursor. Moreover, there are possibly more minor or marginal plant precursors. Confirmation of birch presence requires further research with additional techniques such as GC-MS [25].

The possibility of non-*Populus* plant precursors suggested also caffeoylmalic acid (phaseolic acid) isomer and some unidentified components, especially described in Table 2. They were strongly presented in the mainly two samples (Kogaly and Almaty-1) which may suggest additional unknown plant precursors.

In the end, it is worth adding that not all components presented in propolis have natural origin. Sometimes, there may be traces of different beekeeping techniques (e.g., treatment of honeybees to American and the European foulbrood [29]) or even pollutants occurring in the environment [30]. For this reason, the presence of an uncommon component in propolis should be analyzed in detail.

### 2.2. Comparative Analysis of Chemical Composition of Extracts for Kazakh Propolis Samples

The results of comparative analysis of chemical composition are presented in Figure 2. An investigation based on the uHPLC-DAD matrix (Table 3) with the spectral properties of polyphenols allowed to group of the samples into three main clusters.

The first cluster was composed of six propolis samples with a high presence of different prenyl (methylbutenyl) esters of caffeic acid (mainly 2-methyl-2-butenyl ester, 3-methyl-2-butenyl and 3-methyl-3-butenyl) and pinobanksin-3-O-acetate. In this cluster Bozovoe was less similar to the other samples due to a higher concentration of flavonoid aglycones and lower prenyl esters of caffeic acid. Samples in this cluster exhibited a strong presence of *P. nigra* and similar poplars resins. There were possible other plant precursors; however, they are rather minor or even marginal in most cases.

The second cluster contained three samples with strong presence of *p*-coumaric and cinnamic acids and a lower amount of prenyl esters of caffeic acid than cluster 1. Moreover, in this cluster, there were also some of hydroxycinnamic acid glycerides present. Generally, this cluster represented mixed, aspen–poplar type of propolis.

The last cluster (3) had only one sample (Almaty-7). Its main components were *p*-coumaric, 2-acetyl-1,3-di-*p*-coumaroyl glycerol and *p*-coumaric acid benzyl ester. The composition of Almaty-7 suggested a strong aspen origin with lower amount of poplar markers.

In summary clusters presented in dendrogram reflected presence of poplar and aspen markers described in Section 2.1. Generally, presence of unidentified components in some samples (Kogaly and Almaty 1, 2 and 6) exhibited rather low impact on their clusters grouping—all these samples were presented in two main clusters. For this reason, we may suspect that *Populus* genus was main plant precursor of these samples.

### 2.3. Antimicrobial Activity of Propolis Samples from Kazakhstan

The results of the antimicrobial assays are presented in Table 4. The main goal of our study was a general screening of antimicrobial properties of propolis from Kazakhstan. In our study, we evaluated activity of 70EEP against the following reference microorganisms: six strains of Gram-positive and six strains of Gram-negative bacteria as well as three strains of yeasts.

Among Gram-positive bacteria, Kazakh propolis samples were more active against *Micrococcus luteus* with MIC values of 31.3 µg/mL; however, some samples also showed weak activity (>4000 µg/mL). According to criterium of bioactivity presented by O’Donnell [31], it is defined as good activity. Against two strains of *Staphylococcus aureus* tested, 70EEPs exerted activity 62.5 – > 4000 µg/mL. It is worth mentioning that most of propolis samples presented good bioactivity (62.5–125 µg/mL), and only samples from Kogaly were inactive. Distinguished from the rest of the samples, propolis from Kogaly had its own specific markers in uHPLC-DAD (Table 3) analysis which suggested the presence of an additional unknown plant precursor. It is possible, that its presence may cause strongly lower activity of this sample.

Antibacterial activity of Kazakh propolis against *Staphylococcus epidermidis* and *Bacillus cereus* can be described as good as 31.3–125 µg/mL and 62.5–125 µg/mL, respectively, except the samples obtained from Kogaly, which were inactive (MIC > 4000 µg/mL). Against *Enterococcus faecalis*, propolis from Kazakhstan expressed good or moderate antibacterial activity (62.5–250 µg/mL), which deserves attention. Gram-negative bacteria were insensitive to propolis collected in Kazakhstan (MIC = 4000 µg/mL or more). This may be related to the structure of the bacterial cells and the double cell membrane, which, when exposed to a fraction of surface-active compounds, can stiffen and remodel, increasing its resistance [32]. The presence of the periplasmic space may also cause compounds that have already penetrated the cell to be cut by hydrolytic enzymes, losing their activity [33]. *H. pylori* was the only exception of the Gram-negative bacteria which was inhibited by all propolis samples showing good antibacterial activity against this pathogen (MIC = 31.3–62.5 µg/mL). This high activity can often be associated with impaired urease activity and may affect stick adhesion and cell viability [34]. Moreover, MIC values in all 70EEPs were equal to MBCs. The results presented by the other authors [18,19,35], as well as our own research, clearly indicate that Gram-positive bacteria are susceptible to lower propolis concentrations than Gram-negative ones. *S. aureus* and *B. cereus* are well-known because of their involvement in the gastrointestinal and respiratory tract diseases [15]. Since propolis is usually administered orally, its antimicrobial activity against these pathogens is of great practical importance in its possible therapeutic use [15].

The results of this study showed higher activity of 70EEPs obtained from Kazakhstan in comparison to green and brown propolis ethanolic extracts from Brazil (MIC = 125 and 250 µg/mL, respectively) [15]. Interestingly, partitioning in dichloromethane has enhanced the extraction of antibacterial compounds from Brazilian propolis samples, as it can be inferred from the lower MIC values observed for green (MIC = 7.8 µg/mL) and brown propolis (MIC = 62.5 µg/mL), what was correlated with enhanced levels of phenolic compounds in the extracts [15]. Better activity against *M. luteus* was reported for propolis samples collected in Anatolia (Turkey). Four samples from a different locations, and characterized by the presence of flavonoid compounds (pinocembrin, pinostrobin, isalpinin, pinobanksin, quercetin, naringenin, chrysin and galangin) showed MIC values from 4 to 16 µg/mL [36]. The different species of staphylococci are the microorganisms most often used as models for antimicrobial activity of propolis. This is probably due to their high importance for human morbidity. Staphylococci colonize about 30% of humans (usually asymptomatically) and are responsible for a wide spectrum of difficult-to-treat infections (eye inflammation, pneumonia, meningitis and others) [37]. The anti-staphylococcal potential of tested Kazakh propolis samples against the three reference strains: *S. aureus* ATCC 25923, *S. aureus* ATCC 43300 and *S. epidermidis* ATCC 12228 were better (31.3–250 µg/mL) than results obtained by Grecka et al. (32–4096 µg/mL) [37] and our studies concerning Georgian propolis samples (64–512 µg/mL) [20] and similar to our other result presented in paper about antimicrobial activity of poplar-type propolis (10–2500 µg/mL) [18]. It should be underlined that the MIC is equal to the MBC for most of the tested propolis samples from Kazakhstan. Similar antimicrobial properties (similar ranges of MIC values) of the propolis in question correlate with data on the chemical composition of propolis samples. All samples are characterized by a phytochemical profile (flavonoids and derivatives of phenolic acids) indicating different species of *Populus* as the plant precursor of propolis. The biological activity of propolis is related to time (plant source of exudate), harvest time and geographic origin. Due to these considerations, propolis from a particular geographic region should exhibit similar physicochemical characteristics and other properties, e.g., antimicrobial activity. Several independent studies have shown high susceptibility of *S. aureus* and *S. epidermidis* to different types of propolis originating from Brazil. For example, Reguiera et al. revealed the good bioactivity of Brazilian red propolis hydroalcoholic extracts against *S. aureus* ATCC 6538 and clinical isolates with MIC in the range of concentration 64 to 1024 µg/mL [3]. Another study confirmed the antimicrobial efficacy of ethanolic extracts of three different types of propolis from Brazil: brown, red and green against *S. aureus* ATCC 25923 and *S. aureus* ATCC 25923. The lowest MIC values characterized red type of propolis (25–50 µg/mL), higher in green ones (200–400 µg/mL) and highest (lowest antimicrobial activity) in brown propolis (200–800 µg/mL) [38]. In the same study, the authors compared two methods of raw propolis extraction methods: classical low-pressure extraction with ethanol and supercritical fluid extraction (SFE). Taking into consideration the antibacterial activity, the most potent were ethanolic extracts of propolis, characterized by the highest content of phenolic compounds and high values of flavonoids [38]. These findings confirmed the method we used for the extraction of propolis samples from Kazakhstan due to the increased amount of polar compounds in the extract (phenolic acids and their derivatives and flavonoids) that determine antimicrobial activity. Interestingly, anti-staphylococcal activity of Brazilian red propolis was presented by Regueira et al. [3]. Samples of propolis were collected in the rainy and the dry season. Hydroethanolic extracts showed MIC values for *S. aureus* ATCC 6538 and clinical isolate ≥1024 and 101.6 µg/mL as well 512 and 64 µg/mL for the rainy and the dry season samples, respectively [3]. Comparison of two extracts demonstrated two-times higher concentration of phenolic compounds in the dry season sample, which had the crucial influence on the antibacterial activity [3]. The study reported on the inhibitory and bactericidal properties of 39 South African and three propolis samples from Brazil and was conducted by Suleman et al. [1]. Some samples of African propolis displayed substantial antimicrobial activity with MIC and MBC values at a very low level of 6 µg/mL against *S. aureus* ATCC 25923 [1]; however, the remaining samples had weaker bioactivity (24 up to 1563 µg/mL). The main bioactive constituents of propolis were identified as chrysin, pinocembrin, galangin or 3-pinobankin-3-O-acetate, ingredients, that are also found in poplar type of propolis [1]. The activity of tested Kazakh propolis against the rest of Gram-positive bacteria was similar to results presented in the literature and our own studies [18,20,35,36].

Among all tested Gram-negative bacteria, only *H. pylori* was sensitive for 70EEPs form samples collected in Kazakhstan. To our best knowledge, it is the first communication on the anti-*Helicobacter* activity of Kazakh propolis (except two papers reported one propolis sample from Kazakhstan from different origin). Our group assessed ten samples of 70EEP obtained from different propolis samples derived from various parts of Kazakhstan. The highest bioactivity (31.3 µg/mL) against reference *H. pylori* strain was expressed by 70EEP from Kogaly, Bozove, and four samples from Almaty (Almaty-2, Almaty-3, Almaty-5 and Almaty-6). The rest of samples were characterized by MIC values 62.5 µg/mL. Generally, the antibacterial activity of Kazakh propolis against *H. pylori* and, according to O’Donell criterium, is regarded as good [31]. Moreover, the PE from the sample obtained in Kolgaly has one of the highest activities, despite its inactivity against all Gram-positive bacteria. Surprisingly, for all 70EEPs evaluated against *H. pylori*, MBC/MIC ratio was 1, which confirmed the bactericidal activity of tested propolis extracts [39]. In the frame of our studies, we tried to combine the results of the microbiological evaluation of the inhibition of *H. pylori* growth by propolis extracts from several locations in Kazakhstan with qualitative analysis of their composition by using chromatographic and spectral analysis (uHPLC-DAD and uHPLC-MS/MS). The antibacterial activity of Kazakh 70EEPs were similar to our previous studies focused on the activity propolis from Georgia (MIC = 31.3–125 µg/mL) [40] or different European propolis samples (MIC = 20–30 µg/mL) [18]. Similar results were obtained in the study performed by Santiago et al. [41] with hydroalcoholic extracts of Brazilian propolis against *H. pylori* ATCC 43526 (MIC = MBC = 50.0 µg/mL) and clinical isolate of *H. pylori* (MIC = MBC = 100.0 µg/mL) [41]. The weaker activity against *H. pylori* was presented by 19 propolis samples from Northern Spain (Basque Country) extracted with ethanol and propylene glycol (MIC from 6 to 14 mg/mL) [42]. Indonesian propolis produced by a stingless bee, belonging to *Trigona* spp. was tested against ten clinical isolates of *H. pylori* (from dyspeptic patients) [43]. The results of experiments indicate very weak activity of ethanolic extracts of propolis (MIC = 1024–8192); however, there were promising results of an additive effect against *H. pylori* when used together with clarithromycin and metronidazole [43]. The chemical composition of tested Kazakh propolis, showed a similar phytochemical profile, which is a good explanation of the activity of 70EEPs from Kazakhstan against *H. pylori*. However, results presented by Romero and coauthors showed that activity of complex natural mixtures as propolis is more than just simple sum properties of all constituents and interaction among them, which should be taken into consideration [44]. Finally, analyses by transmission electron microscopy at sub-inhibitory concentration showed vesicle formation and bacterial cell lysis after exposition to individual polyphenols and in the mixture, suggesting a potential bactericidal activity of propolis [44].

Antifungal activity of 70EEPs was tested against three *Candida* species. Propolis exhibited moderate antifungal activity (125–500 µg/mL) against *C. glabrata* and moderate-to-mild bioactivity (125–1000 µg/mL) against *C. albicans* and *C. parapsilosis*. There are numerous experimental works on the antimicrobial activity of various kinds of propolis collected from different geographical locations [1,15,36,45]. Depending on the content of the samples, propolis may inhibit the process of filamentation and yeast adhesion and increase intracellular oxidative stress [46]. The bioactivity of tested samples against pathogenic yeasts (62.5–1000 µg/mL) is weaker than our research concerning propolis for samples obtained from Georgia and central Europe, but similar to the activity of propolis samples from South Africa (MIC between 98–1563 µg/mL] [1] and Cretan propolis (370–1560 µg/mL) [45]. Better activity against *C. albicans* was exerted by Anatolian propolis (MIC range 4–32 µg/mL) [36]. Propolis owes its antimicrobial activity mainly to the presence of polyphenolic compounds (phenolic acids and flavonoids).

The mechanism underlying the antimicrobial activity of propolis involves the flavonoid and phenolic acids present in propolis. The literature data, among them some reviews confirmed and summarized these mechanisms of activity [3]. Polyphenols are responsible for the inhibition of nucleic acid synthesis (DNA and RNA) and the inhibitory mechanism on DNA gyrase (procaryotic enzyme plays an important role in processes of replication, transcription and recombination) [47,48,49]. Galangin (flavonol) and derivatives of caffeic acids derivatives (polyphenolic acids) have the ability to uncouple the energy-transducing cytoplasmic membrane and inhibit bacterial motility. Moreover, these effects on the bioenergetic status of the membrane may contribute to the antimicrobial action of propolis and its observed synergism with selected antibiotics [50]. Flavonoids, among other phenolic compounds, interfere with the energy metabolism of the bacterial cell due to the damage to the cytoplasmatic membranes, their permeability alteration and the perturbance in the exchange of nutrients and metabolites [35,47,48,49]. Additionally, flavonoids from propolis inhibit adhesion and biofilm formation [3,35,47,48,49].

In summary observed differences of the antimicrobial activity of Kazakh propolis may bresultesult of different factors. The basic one may be differences between Kazakh propolis plant precursors. Usually, propolis with a stronger presence of poplar markers is expected to be stronger than propolis with aspen–poplar origin [18,40]. However, *P. treumula* exudates sometimes exhibited stronger activity than *P. nigra* resins [51,52,53]. Moreover, the same propolis may exhibit lower or stronger activity than its plant precursor [51,52,53]. Propolis as well as *Populus* genus bud exudates are a complex matrix and their antimicrobial activity is an effect of interaction between many components. The same exudates of this same *Populus* species may be observed in different chemotypes [26,51,52] that should also exhibit an impact on propolis antimicrobial activity. In results, total effect may also be connected with presence of minor and even marginal plant precursors as well as specific chemical composition of plant precursor.

### 2.4. Urease Inhibitory Activity and Anti-Helicobacter Activity of Tested Kazakh Propolis Extracts

The results for the assessment of selected 70EEPs from Kazakh propolis are listed in Table 5.

The presented study is the first attempt to evaluate the effects of 70EEPs obtained from propolis sample collected in Kazakhstan, a natural bee product used in the treatment of gastric diseases, on *H. pylori* growth in vitro as well as the activity of its enzyme urease which is crucial for the ability of the pathogen to colonize the stomach. Results of bioassay shows IC_50_ values for 70EEPs ranging from 440.73 to 11,177.24 µg/mL and IC_50_ = 92.7 µg/mL for thiourea (reference inhibitor) (Table 5 and Figure 3). Obtained results suggested the same plant origin of Kazakhstan propolis. Presented results suggested that inhibition of urease is not directly connected with bactericidal activity against *H. pylori*. Moreover, variability of activity inside clusters also suggested that the same type of propolis may not be directly connected with urease inhibition activity. It is more possible that the obtained effect is a result of interaction between components in complex natural matrix of propolis. For this reason, some samples may exhibit better activity inside this same cluster.

Results showed in described experiments are similar to the other research of 70EEPs inhibitory activity and indicate that searching for a novel, natural urease inhibitors among bee products is the proper direction. For example, Baltsas et al. [54] tested 15 PEs of Turkish propolis samples for urease inhibitory activity. The tested propolis samples exerted IC_50_ in the range of 0.260 to 1.525 mg per mL, similar to the results exhibited in this study. Inhibition activity of 70EEPs from Kazakhstan is distinctly weaker than other natural products, e.g., essential oils. For example, essential oil from *Origanum vulgare* (MIC = 31.3 µg/mL) presented IC_50_ against *H. pylori* urease equal to 208.3 µg/mL [55]. Moreover, the most active essential oil (cedarwood essential oil) has IC_50_ = 5.3 µg/mL (MIC = 15.6 µg/mL) [55].

In research performed by Can [56], 11 propolis samples from the Marmara region of Turkey were tested. Their activity concerning urease inhibition was in the range from 1.110 to 5.870 mg/mL and authors suggested that it indicates good bioactivity of tested propolis extracts [56]. In another experiment done by Can [57], where enzyme inhibition of urease was examined by different bee products—honey, pollen and propolis. The IC_50_ values were changed from 7.02 to 33.25 mg/mL, 5.00 to 8.78 and 0.16 to 1.98 mg/mL in the honey, pollen and propolis samples, respectively [57].

Urease is a crucial enzyme for *H. pylori* to survive in an acidic environment of the stomach. Propolis extracts, which contain numerous polyphenolic compounds that have the ability to inhibit urease, can be considered a useful component of *H. pylori* eradication therapy.

## 3. Materials and Methods

### 3.1. Propolis and Chemicals

Propolis samples from the following region of Kazakhstan were collected in 2021: Almaty region (7 samples), Kogaly, Kegen and Bozone. Obtained raw propolis samples were frozen in liquid nitrogen and crushed in a mortar. Procedures were repeated three times. Before extraction, ground propolis was stored in sealed containers under −20 °C.

LiChrosolv^®^ hyper grade eluents for uHPLC-MS/MS and uHPLC-DAD analysis (acetonitrile, water and methanol) were purchased from Merck company (Darmstad, Germany). Mueller-Hinton agar and Sabouraud agar were obtained from Oxoid (Hampshire, UK).

### 3.2. Preparation of Propolis Extracts (70EEPs)

Grounded research material was extracted by ethanol in water (70:30; *v*/*v*) in proportion 1:10 (1.0 g of propolis sample per 10 mL of solution). Extraction was performed in an ultrasonic bath (Sonorex, Bandelin, Berlin, Germany). Extraction conditions were set on 20 °C for 45 min and 756 W (90% of ultrasound bath power). Obtained extracts were stored at room temperature for 12 h and then filtered through Whatman No. 10 paper (Cytiva, Marlborough, MA, USA).

### 3.3. UHPLC-DAD-MS/MS Profile of Propolis Extracts

uHPLC analyses were performed as the previously described [20] with a Thermo Scientific UltiMate 3000 system (Thermo Scientific™ Dionex™, Sunnyvale, CA, USA), coupled with an autosampler and DAD detector recording spectral data in the 200–600 nm range and monitoring at 280, 320 and 360 nm. uHPLC-MS/MS was carried out using Compact QqTOF MS/MS detector (Bruker, Darmstadt, Germany). MS detector was used in electrospray negative mode. Conditions of analysis were: ion source temperature was set to210 °C, nebulizer gas pressure to 2.0 bar, dry gas (nitrogen) flow 8.01 L/min and temperature to 210 °C. The capillary voltage was programmed to 4.5 kV. The collision energy was set to 8.0 eV. Internal calibration was obtained with a 10 mM solution of sodium formate. For ESI-MS/MS experiments, collision energy was set at 35.0 eV and nitrogen was used as collision gas. Scan range was set from 30 to 1300 *m*/*z*.

### 3.4. Determination of Antimicrobial Activity

The propolis extracts dissolved in dimethylo-sulfoxide (DMSO) were screened for antibacterial and antifungal activities by microdilution broth method according to both the European Committee on Antimicrobial Susceptibility Testing (EUCAST) (www.eucast.org (accessed on 3 January 2023) using Mueller–Hinton broth or RPMI with MOPS for growth of fungi as we described elsewhere [58,59]. Minimal inhibitory concentration (MIC) of the tested extracts were evaluated for the wide panel of the reference microorganisms, including Gram-negative bacteria (*Escherichia coli* ATCC 25922, *Salmonella typhimurium* ATCC14028, *Klebsiella pneumoniae* ATCC 13883, *Pseudomonas aeruginosa* ATCC 9027), Gram-positive bacteria (*Staphylococcus aureus* ATCC 25923, *Staphylococcus aureus* ATCC 43300, *Staphylococcus epidermidis* ATCC 12228, *Micrococcus luteus* ATCC 10240, *Enterococcus faecalis* ATCC 29212, *Bacillus cereus* ATCC 10876) and fungi (*Candida albicans* ATCC 10231, *Candida parapsilosis* ATCC 22019, *Candida glabrata* ATCC 90030). The sterile 96-well polystyrene microtitrate plates (Nunc, Roskilde, Denmark) were prepared by dispensing 100 µL of appropriate dilution of the tested extracts in broth medium per well by serial two-fold dilutions in order to obtain final concentrations of the tested extracts ranged from 0.0195 to 10 mg/mL The inocula were prepared with fresh microbial cultures in sterile 0.85% NaCl to match the turbidity of 0.5 McFarland standard were added to wells to obtain final density of 5 × 10^5^ CFU/mL for bacteria and 5 × 10^4^ CFU/mL for yeasts (CFU, colony forming units). After incubation (35 °C for 24 h), the MICs were assessed visually as the lowest concentration of the extracts showing complete growth inhibition of the reference microbial strains. Appropriate DMSO control (at a final concentration of 10%), a positive control (containing inoculum without the tested derivatives) and negative control (containing the tested derivatives without inoculum) were included on each microplate.

The MIC for *H. pylori* ATCC 43504 was determined using a two-fold microdilution method in MH broth with 7% of lysed horse blood at extract concentration ranging from 1000 to 1.95 mg/L with bacterial inocula of 3 McFarland standard. After incubation at 35 °C for 72 h under microaerophilic conditions (5% O_2_, 15% CO_2,_ and 80% N_2_) the growth of *H. pylori* was visualized with the addition 10 µL of 0.04% resazurin. The MIC endpoint was recorded after 4 h incubation as the lowest concentration of extract that completely inhibits growth [55].

Minimal bactericidal concentration (MBC) or minimal fungicidal concentration (MFC) was obtained by culture of 5 mL from each well that showed through growth inhibition, from the last positive one, and from the growth control onto recommended agar plates. The plates were incubated at 35° for 24 h for all microorganisms but *H. pylori* which were incubated for 72 h in microaerophilic conditions.

The MBC/MFC was defined as the lowest concentration of extract without the growth of microorganisms. The MBC/MIC ratios were calculated to determine the bactericidal or bacteriostatic effect of the assayed extract. Vancomycin, clarithromycin, ciprofloxacin and nystatin were used as the reference drugs appropriate for different group of microorganisms.

The experiments were repeated in triplicate. Representative data are presented.

### 3.5. Urease Inhibitory Assay

In short, *H. pylori* were incubated for 72 h in the MH broth with 7% of horse serum (Sigma-Merk, Saint Louis, Missouri, USA) in microaerophilic conditions. Bacterial biomass was collected by centrifugation at 4000× *g* at 4 °C for 10 min, then the cells were dissolved in ice-cold phosphate buffer (pH 7.3) with a protease inhibitor cocktail (Sigma). The urease enzyme was prepared by disturbing *H. pylori* cells by sonication, followed by centrifugation at 12,000× *g* at 4 °C for 10 min.

Initial urease inhibitory activity of all the obtained extracts were evaluated at the concentration of 2 mg/mL with the modified Berthelot spectrophotometric method with phenol–hypochlorite reaction at the absorbance of 570 nm. The enzyme reaction was activated in 96-well plates by mixing the appropriate volume of 2% urea, sodium phosphate buffer solution (100 µL), different concentrations (2000–3.9 µg/mL) of propolis extract, and the reaction mixture was incubated for 15 min at 37 °C, then the concentration of ammonia was determined using the Berthelot method. The amount of the ammonia is equivalent to the hydrolysis of urea using the urease enzyme. The experiments were performed in triplicate. Activity of uninhibited urease was chosen as the control activity of 100% [60]. Inhibition rate (%) was calculated following the formula: I% = (1 − average with inhibitors/average activity without inhibitors) × 100%. The IC_50_ was expressed as the concentration of inhibitor that decreased urease activity by 50% and calculated by plotting the percent of inhibition using the internet IC_50_ Calculator (AAT Bioquest).

### 3.6. Statistical Analysis

Statistical analysis was performed by Statistica 14.0.0.5 software (Tibco Sofware Inc., Palo Alto, CA, USA). Analysis included hierarchical fuzzy clustering trees (dendrogram) from the prepared matrix. It was composed of % of UV chromatograms (280 nm) relatively peak area. Substances about at least 1% of relatively area (in any sample) were used to construct matrix.

## 4. Conclusions

The phytochemical profile and activity against 15 microorganisms 70EEP from ten propolis samples collected in Kazakhstan were evaluated. This is the first wider study on Kazakh propolis extracts phytochemical composition and the antimicrobial potential. Tested extracts exhibited good activity against Gram-positive bacteria, fungal species (yeasts) and *H. pylori* (the only Gram-negative bacterium sensitive to the tested propolis). In addition, bioactivity tests were conducted for urease inhibition. Propolis from Kazakhstan seems to belong to the poplar type, but analysis of the chemical composition showed the presence of polyphenolic compounds from other plant sources (especially aspen) which requires further research. Dependences between their plant origin and activity was ambiguous. This may be caused be specific chemotype of Kazakstani *Populus* species or presence additional, unknown plant precursors. An attempt of the isolation the active components from tested propolis samples should be performed in the future, in order to study more the origin of propolis and its various biological activities.

## Figures and Tables

**Figure 1 molecules-28-02984-f001:**
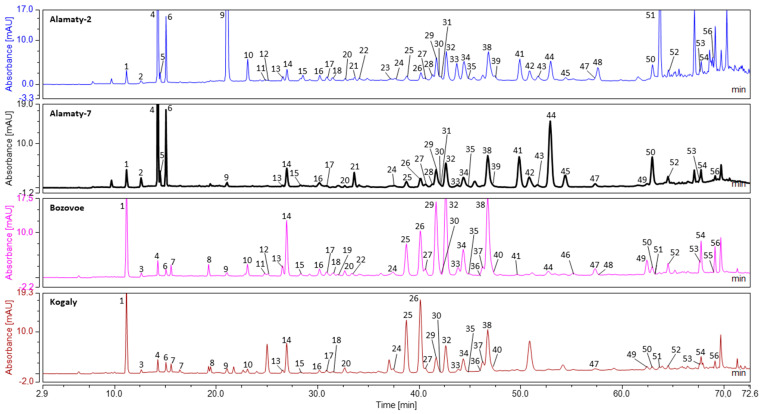
Representative uHPLC-DAD chromatograms of Kazakhstan propolis. **Figure Legend:** 1—Caffeic acid; 2—Vanilline; 3—Caffeoylglycerol; 4—*p*-Coumaric acid; 5—Benzoic acid; 6—Ferulic acid; 7—Isoferulic acid; 8—Caffeoylmalic acid (Phaseolic acid) isomer; 9—Cinnamic acid; 10—Pinobanksin-5-methyl ether; 11—Quercetin; 12—luteloin; 13—Quercetin-3-methyl-ether; 14—Pinobanksin; 15—Naringenin; 16—Apigenin; 17—Kaempferol; 18—Isorhamnetin; 19—Quercetin-methyl-ether; 20—Luteolin-5-methyl-ether; 21—1,3-di-*p*-Coumaroylglycerol; 22—Quercetin-dimethyl-ether; 23—2-Acetyl-1,3-di-caffeoylglycerol; 24—Rhamnetin; 25—Caffeic acid 2-methyl-2-butenyl ester; 26—Caffeic acid 3-methyl-2-butenyl ester; 27—Caffeic acid 3-methyl-3-butenyl ester; 28—(R/S) 2-Acetyl-1-caffeoyl-3-*p*-coumaroylglycerol; 29—Chrysin; 30—Caffeic acid benzyl ester; 31—(R/S) 2-Acetyl-1-caffeoyl-3-feruloylglycerol; 32—Pinocembrin; 33—Sakuranetin; 34—Galangin; 35—Genkwanin; 36—Caffeic acid pentyl or isopentylester ester; 37—Caffeic acid phenethyl ester (CAPE); 38—Pinobanksin 3-O-acetate; 39—Kaempferide (Kaempferol 4′-methyl ether); 40—Methoxychrysin; 41—2-Acetyl-1,3-di-p-coumaroylglycerol; 42—(R/S) 2-Acetyl-3-*p*-coumaroyl-1-feruloylglycerol; 43—2-Acetyl-1,3-di-feruloylglycerol; 44—*p*-Coumaric acid benzyl ester; 45—Ferulic acid benzyl ester; 46—Caffeic acid cinnamyl ester; 47—Pinobanksin-3-O-propanoate; 48—*p*-Coumaric acid phenethyl ester; 49—Tectochrysin; 50—Pinostrobin; 51—*p*-Coumaric acid cinnamyl ester; 52—Pinobanksin 3-O-butanoate or isobutanoate; 53—Pinobanksin 3-O-pentanoate or isopentenoate isomer I; 54—Pinobanksin 3-O-pentanoate or isopentenoate isomer II; 55—Pinobanksin-3-O-hydroxycinnamate; 56—*p*-Methoxy cinnamic acid cinnamyl ester.

**Figure 2 molecules-28-02984-f002:**
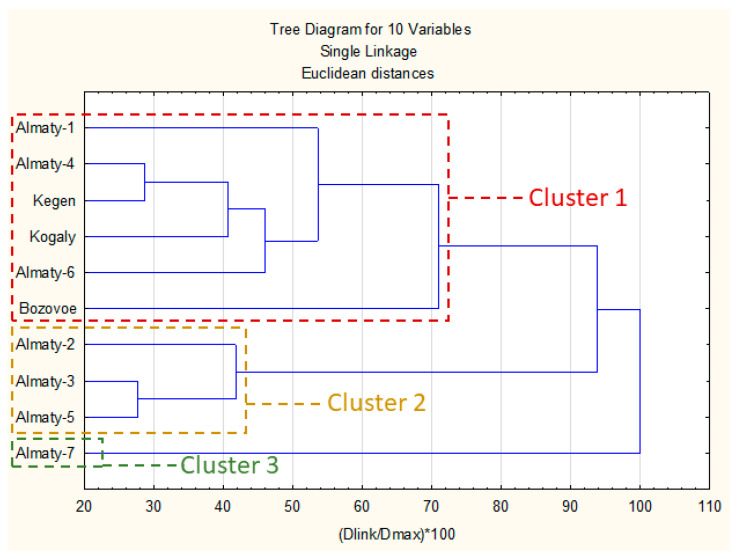
Dendrogram of Kazakh propolis chemical composition. **Figure legend:** Dlink—Linkage Distance; D max—Maximal distance.

**Figure 3 molecules-28-02984-f003:**
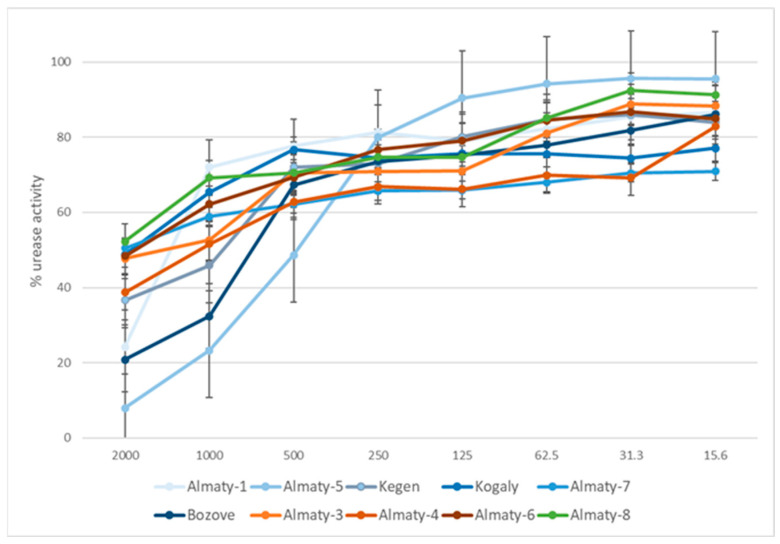
Urease activity inhibition by tested 70EEP of Kazakh propolis.

**Table 1 molecules-28-02984-t001:** Identification of propolis components in 70EEP by uHPLC-MS/MS analysis.

No.	Component	RT MS	[M-H^+^]^−^	Base MS/MS Peak *m*/*z*	Secondary MS/MS Peaks *m*/*z* (A [%])	[M-H^+^]^−^ Formula	Error [mDa]	Error [ppm]	RDB
1	Unidentified	0.88	179.0565	-	-	C6H11O6	−0.4	−2.2	1
2	Unidentified	1.01	133.0144	-	-	C4H5O5	−0.2	−1.2	2.0
3	Unidentified	1.24	167.0210	-	-	C4H7O7	−1.3	−7.8	1.0
4	Unidentified	1.43	117.0189	-	-	C4H5O4	0.3	3.8	2.0
5	4-Hydroxybenzoic acid ^a,b,c^	6.57	137.0246	-	-	C7H5O3	−0.1	−1.1	5.0
6	Unidentified	8.01	137.0246	-	-	C7H5O3	−0.2	−1.4	5.0
7	Vanillin isomer ^b,c^	9.33	151.0393	108.2066	-	C8H7O3	0.8	5.2	5.0
8	Unidentified	9.84	121.0293	-	-	C7H5O2	0.2	1.7	5.0
9	Unidentified	10.43	357.1197	195.1661	-	C16H21O9	−0.6	−1.6	6.0
10	* Caffeoylglycerol ^b,c^	10.53	253.0717	133.1749	161.0724 (75.43)	C12H13O6	0.1	0.2	6.0
11	* Catehin or Epicatehin ^b,c^	10.83	289.0728	137.6817	203.2588 (63.79)	C15H13O6	−0.2	−0.8	9.0
12	Unidentified	10.87	177.0192	-	-	C9H5O4	0.1	0.7	10.0
13	Unidentified	10.91	165.0557	-	-	C9H9O3	0.0	0.3	5.0
14	Caffeic acid ^a,b,c^	11.45	179.0346	135.0449	107.0484 (8)	C9H7O4	0.4	2.0	6.0
15	* Caffeoylglycerol ^b,c^	13.03	253.0711	161.0743	133.1839 (92.59), 135.1153 (40.05)	C12H13O6	0.6	2.5	6.0
16	Unidentified	14.06	195.0663	121.1371	-	C10H11O4	0.0	0.2	5.0
17	* Caffeic acid dihydroxypentyl or isopentyl ester isomer I ^b,c^	14.33	281.1036	161.1260	133.7160 (76.42)	C14H17O6	−0.5	−1.9	6.0
18	Unidentified	14.35	165.0194	-	-	C8H5O4	−0.1	−0.6	6.0
19	Unidentified	14.39	237.0770	145.1304	117.1516 (97.17)	C12H13O5	−0.2	−0.8	6.0
20	*p*-Coumaric acid ^a,b,c^	14.41	163.0401	119.1668	93.0893 (10.59)	C9H7O3	0.0	−0.1	6.0
22	* Caffeic acid dihydroxypentyl or isopentyl ester isomer II ^b,c^	14.74	281.1034	161.1323	133.1404 (55.05), 135.1467 (28.11)	C14H17O6	−0.3	−1.0	6.0
23	Ferulic acid ^a,b,c^	15.19	193.0504	134.1169	-	C10H9O4	0.2	1.2	6.0
24	* Caffeic acid dihydroxypentyl or isopentyl ester isomer III ^b,c^	15.22	281.1033	161.1496	133.2606 (51.72), 135.1486 (42.76), 179.1248 (10.81)	C14H17O6	−0.3	−1.0	6.0
25	Isoferulic ^a,b,c^	15.7	193.0503	134.1466	-	C10H9O4	0.0	−0.2	6.0
26	Unidentified	15.93	205.0509	119.1273	-	C11H9O4	−0.3	−1.2	7.0
27	* Caffeoylmalic acid (Phaseolic acid) isomer ^b,c^	16.64	295.0827	161.1286	133.2853 (58.51), 135.1555 (32.02)	C14H15O7	−0.3	−1.1	7.0
28	Unidentified	17.75	193.0505	133.1880	-	C10H9O4	0.1	0.2	6.0
29	Unidentified	18.55	359.1137	145.1441	119.1324 (55.67), 163.1822 (40.34), 117.2787 (26.02), 153.1744 (9.79), 150.2477 (5.81), 165.2462 (5.36)	C19H19O7	0.0	−0.1	10.0
30	Eriodyctiol (4′-hydroxynaringenin) ^b,c^	18.82	287.0562	125.0569	177.1795 (70.51), 201.1649 (12.51), 259.2324 (12.01), 213.2079 (9.60), 241.2596 (8.96), 131.2132 (7.78)	C15H11O6	−0.1	−0.4	10.0
31	Unidentified	19.63	279.0875	145.1378	117.1459 (61.44), 119.1462 (23.20)	C14H15O6	−0.1	−0.5	7.0
32	Apigetrin ^b,c^	21.06	431.0983	268.2682	431.2804 (23.37), 240.1429 (9.85), 211.1568 (9.64)	C21H19O10	0.0	0.1	12.0
33	Cinnamic acid ^a,b,c^	21.21	147.0450	-	-	C9H7O2	0.2	1.2	6
34	Unidentified	21.37	263.0921	133.6382	161.1261 (74.66)	C14H15O5	0.4	1.7	7
35	Unidentified	21.61	285.0778	138.1476	224.1795 (91.95), 252.3284 (54.53), 239.2369 (42.40), 197.2624 (24.65)	C16H13O5	−1.0	−3.4	10.0
36	Unidentified	21.94	349.0931	201.1446	-	C17H17O8	−0.2	−0.5	9.0
37	Caffeic acid derivate ^b,c^	22.59	207.0664	133.2876	135.1197 (48.56), 161.1252 (19.75)	C11H11O4	−0.2	−0.8	6
38	Unidentified	23.01	287.0560	135.1477	-	C15H11O6	0.1	0.4	10
39	Pinobanksin 5-methylether ^b,c^	23.32	285.0777	252.0429	224.0470 (55.83), 138.0332 (38.07), 241.0481 (31.50), 165.0192 (14.95), 239.0674 (12.13), 195.0459 (12.02), 151.0027 (11.81), 213.0557 (11.34), 267.0660 (11.02), 285.0805 (9.31), 136.0190 (8.53), 107.0176 (6.81)	C16H13O5	−0.8	−2.9	10.0
40	Unidentified	23.40	277.1084	161.1236	135.1475 (19.77), 179.1557 (6.83), 277.3681 (6.48)	C15H17O5			
41	Unidentified	24.05	277.1088	161.1087	135.1097 (29.04), 277.2969 (13.31)				
42	Unidentified	24.56	315.0872	282.2485	267.2232 (91.67), 239.2026 (65.45), 138.3330 (63.14), 165.1889 (39.96), 271.2613 (32.25)	C17H15O6	0.2	0.7	10.0
43	Quercetin ^a,b,c^	25.05	301.0353	151.0034	121.0307 (29.41), 107.0140 (22.18), 149.0242 (14.01), 178.9969 (13.92), 301.0371 (7.58), 245.0461 (6.32), 273.0451 (5.48), 163.0034 (4.87), 211.0372 (3.84)	C15H9O7	0.1	0.3	11.0
44	Luteolin ^a,b,c^	25.38	285.0412	133.1356	285.1812 (83.77), 151.0369 (33.21), 199.1521 (15.09), 107.1489 (12.83)	C15H9O6	−0.8	−2.7	11.0
45	Quercetin 3-methyl ether ^b,c^	26.82	315.0497	271.0253	300.0274 (71.14), 255.0303 (42.89) 243.0297 (22.59), 227.0334 (2.55)	C16H11O7	0.2	0.5	11.0
46	Pinobanksin ^a,b,c^	27.16	271.0615	197.0617	253.0502 (89.28), 161.0604 (67.51), 271.0605 (56.26), 125.0242 (53.39), 151.0063 (30.14), 225.0558 (24.71), 107.0152 (23.97), 209.0588 (16.07), 185.0571 (15.86), 115.0559 (15.08), 157.0659 (14.43), 181.0651 (14.14), 215.0699 (11.83)	C15H11O5	−0.3	−1.1	10.0
47	Naringenin ^a,b,c^	28.55	271.0612	119.1344	151.0545 (43.37), 107.0883 (21.94), 187.2234 (10.00)	C15H11O5	0.0	0.1	10.0
48	Chrysin-5-methyl-ether ^b,c^	28.75	267.0662	224.1747	180.1680 (92.97), 252.1932 (26.27), 195.2896 (15.00)	C16H11O4	0.1	0.3	11.0
49	Unidentified	28.76	301.7210	152.1363	301.2899 (54.11), 283.2605 (49.47), 125.0784 (41.62), 176.1333 (40.92), 227.2018 (25.22), 268.2906 (23.80), 212.2146 (17.95), 107.2091 (14.12), 240.2041 (12.08), 191.1941 (10.43), 224.1551 (9.53), 255.2235 (9.08), 172.2100 (8.91), 165.1410 (8.49), 180.1334 (7.67), 200.6049 (9.89), 196.6155 (8.14), 245.2481 (5.47),	C16H13O6	−0.3	−1.1	10.0
50	1-Caffeoyl-3-p-coumaroylglycerol ^b,c^	28.89	399.1085	163.1721	161.0857 (48.44), 119.1488 (48.96), 253.2139 (46.08), 179.1589 (25.62), 145.1790 (24.73), 235.1152 (20.40), 161.2192 (10.73), 237.2187 (8.31), 399.2525 (5.30)	C21H19O8	0	0.1	12.0
51	Unidentified	29.48	269.0822	150.0692	184.1621 (88.87), 165.1076 (80.74), 122.0565 (55.22), 254.1667 (50.90), 227.1995 (38.24), 269.26 (20.13)	C16H13O4	−0.3	−1	10.0
52	Caffeic acid propyl or isopropyl ester ^b,c^	29.8	221.0826	133.7159	161.1045 (20.87)	C12H13O4	−0.6	−2.8	6.0
53	Unidentified	30.21	301.0717	164.0930	151.1346 (92.98), 136.0892 (53.24)	C16H13O6	−0.1	−0.2	10.0
54	Unidentified	30.39	389.1977	137.1138	389.7861 (95.55)	C22H29O6	−0.7	−1.9	8.0
55	Apigenin ^a,b,c^	30.47	269.0457	117.0349	269.0455 (52.06), 151.0033 (39.01), 149.0245 (25.91), 227.0353 (12.66), 107.0138 (11.48), 225.0555 (10.59), 201.0561 (7.44), 183.0448 (6.40), 181.0630 (5.14), 121.0290 (4.92), 197.0608 (2.28)	C15H9O5	−0.2	−0.7	11.0
56	Kaempferol ^a,b,c^	31.14	285.0405	285.0400	239.0335 (8.81), 187.0408 (8.20), 185.0580 (8.14), 229.0505 (7.99), 159.0464 (6.63)	C15H9O6	−0.1	−0.3	11.0
57	Isorhamnetin ^a,b,c^	31.72	315.0509	300.1989	151.1329 (26.66), 271.4108 (11.37), 164.1072 (7.61), 283.1502 (6.12), 148.0893 (5.64), 315.1957 (5.60), 255.2267 (4.65), 216.1788 (3.38), 108.2193 (2.95), 244.2404 (2.60), 136.2082 (2.55)	C16H11O7	0.1	0.3	11.0
58	Unidentified	31.86	387.1814	137.1045	-	C22H27O6	−0.1	−0.3	9.0
59	Quercetin-methyl-ether ^b,c^	32.23	315.0511	300.1857	151.1387 (26.12), 271.2935 (11.15), 164.1172 (7.58), 283.1466 (5.81), 216.2658 (4.63)	C16H11O7	0.0	−0.1	11.0
60	Unidentified	32.82	417.1928	167.1338	152.0939 (17.25), 123.1301 (16.41), 108.1042 (7.08)	C23H29O7	−0.9	−2.2	9.0
61	Luteolin-5-methyl ether ^b,c^	32.97	299.0549	255.0300	227.0344 (59.96), 284.0336 (15.07), 211.0379 (6.11)	C16H11O6	−0.2	−0.7	11.0
62	(R/S) 1,2-di-*p*-Coumaroylglycerol isomer I ^b,c^	33.02	383.1137	163.1661	119.1192 (71.11)	C21H19O7	−0.1	−0.3	12.0
63	Caffeic acid buten or isobuten ester ^b,c^	33.60	233.0830	133.3638	-	C13H13O4	−1.1	−4.7	7.0
64	Quercetin-di-methyl-ether ^b,c^	33.68	329.0669	271.1688	299.1957 (99.34), 243.1827 (90.63), 285.4120 (51.12), 257.2245 (31.51), 314.2443 (29.44), 227.1660 (5.23), 215.1776 (3.74), 199.1937 (3.06), 255.1517 (2.88)	C17H13O7	−0.2	−0.6	11.0
65	1,3-di-*p*-Coumaroylglycerol ^b,c^	33.91	383.1143	163.1491	119.1294 (69.49), 145.1419 (61.09), 117.2337 (8.68), 219.1918 (7.20), 237.1927 (6.59), 383.3604 (2.42)	C21H19O7	−0.7	−1.8	12.0
66	Unidentified	33.93	373.2015	373.4715	137.1102 (64.03), 235.2731 (17.95), 149.2226 (9.88)	C22H29O5	0.5	1.4	8.0
67	Unidentified	33.98	359.0777						
68	Unidentified	34.35	391.2134	391.4698	137.0988 (25.55)	C22H31O6	−0.8	−2	7.0
69	Galangin-5-methyl-ether ^b,c^	34.37	283.0612	211.1796	239.2387 (58.94), 283.2956 (5.07), 268.1859 (4.79)	C16H11O5	0.0	−0.1	11.0
70	(R/S) 1-*p*-Coumaroyl-3-feruloylglycerol ^b,c^	34.38	413.1241	193.1678	163.1401 (97.02), 134.1556 (76.61), 119.1270 (54.22), 145.1831 (49.19), 175.1423 (37.15), 149.1613 (18.59), 398.3044 (15.16), 161.2714 (11.03), 413.4833 (10.86), 219.2266 (8.25), 237.2114 (7.99), 249.2240 (7.20), 252.2234 (6.36), 267.1968 (5.71), 235.2153 (5.19)	C22H21O8	0.1	0.2	12.0
71	(R/S) 1,2-di-p-Coumaroylglycerol isomer II ^b,c^	34.39	383.1137	163.1447	119.1053 (78.80), 145.1222 (70.92)	C21H19O7	−0.1	−0.2	12.0
72	5-Methyl-pinobanksin-3-acetate ^b,c^	34.61	327.0878	224.1781	267.2163 (67.46), 252.1858 (62.85), 285.2285 (45.11), 239.5247 (36.67)	C18H15O6	−0.4	−1.1	11.0
73	2-Acetyl-1,3-di-caffeoylglycerol ^b,c^	35.15	457.1141	179.1565	161.1483 (77.42), 135.1105 (45.90), 235.2026 (48.11), 295.2730 (38.65), 457.3254 (5.86), 173.1999 (3.85), 397.3589 (4.20), 413.5593 (3.26), 253.2546 (2.22)	C23H21O10	−0.1	−0.2	13.0
74	Unidentified	36.21	229.0874	174.1260	146.1117 (43.89), 206.1324 (18.98), 213.2651 (14.25), 229.2511 (8.28)	C14H13O3	−0.4	−1.6	8.0
75	Rhamnetin ^a,b,c^	36.48	315.0509	165.1079	121.1282 (39.04), 300.2162 (27.72), 151.1032 (9.49), 272.2119 (6.69), 244.2122 (4.72), 256.2717 (3.45)	C16H11O7	0.1	0.4	11.0
76	Kaempferol-methyl-ether ^b,c^	36.64	299.0563	284.1907	299.2151 (7.35), 256.1440 (5.21), 133.2419 (5.23), 151.0642 (2.37), 227.3301 (2.53)	C16H11O6	−0.2	−0.7	11.0
77	Caffeic acid butyl or isobutyl ester isomer I ^b,c^	37.17	235.0976	161.1424	135.1301 (93.59)	C13H15O4	−0.1	−0.2	6.0
78	Caffeic acid prenyl derivate ^b,c^	37.70	247.0975	135.1279	161.1137 (33.38)	C14H15O4	0.0	0.2	7.0
79	Caffeic acid butyl or isobutyl ester isomer II ^b,c^	37.92	235.0978	133.5359	161.1498 (41.79)	C13H15O4	−0.2	−1	6.0
80	Unidentified	38.17	373.2012	136.1022	92.1596 (18.38), 373.3655 (5.60)	C22H29O5	0.8	2.2	8.0
81	Quercetin-dimethyl-ether ^b,c^	38.75	329.0669	299.1970	271.1734 (30.28), 314.2379 (21.06), 285.2543 (2.46)	C17H13O7	−0.3	−0.8	11.0
82	Caffeic acid 2-methyl-2-butenyl ester ^b,c^	39.04	247.0979	135.1258	161.1463 (36.02), 179.1152 (11.25)	C14H15O4	−0.4	−1.5	7.0
83	Caffeic acid 3-methyl-2-butenyl ester (Basic prenyl ester) ^b,c^	40.44	247.0979	134.2235	106.1200 (6.32)	C14H15O4	−0.4	−1.7	7.0
84	Caffeic acid 3-methyl-3-butenyl ester ^b,c^	40.90	247.0977	134.2234	106.1659 (5.64)	C14H15O4	−0.1	−0.4	7.0
85	(R/S) 2-Acetyl-1-caffeoyl-3-p-coumaroylglycerol ^b,c^	41.69	441.1197	163.1479	179.1479 (85.75), 161.1248 (42.10), 135.1226 (40.85), 145.1602 (39.56), 119.1276 (35.73), 235.2124 (27.59), 295.2823 (14.64), 219.1731 (7.31), 173.1816 (6.88), 381.3956 (7.79), 217.1798 (4.50), 441.3513 (4.75), 189.1920 (3.80), 277.2596 (2.86)	C23H21O9	−0.6	−1.3	13.0
86	Chrysin ^a,b,c^	42.00	253.0505	253.0507	143.0507 (41.53), 145.0299 (21.10), 209.0611 (14.10), 107.0142 (13.33), 181.0652 (8.16), 185.0615 (6.19)	C15H9O4	−0.7	−2.8	11.0
87	Caffeic acid benzyl ester ^b,c^	42.28	269.0818	134.1302	161.0235 (22.96), 137.0256 (4.03)	C16H13O4	−0.3	−1.1	10.0
88	(R/S) 2-Acetyl-1-caffeoyl-3-feruloylglycerol ^b,c^	42.49	471.1297	193.1684	179.1426 (89.35), 161.1376 (39.08), 135.1206 (36.34), 175.1354 (30.55), 235.2142 (27.00), 295.2633 (15.17), 149.1373 (11.76), 411.3719 (10.46), 173.2002 (6.78), 471.4677 (7.40), 249.2085 (5.71), 217.2027 (5.85), 189.2351 (3.58), 277.2277 (3.10), 367.3075 (2.44)	C24H23O10	−0.1	−0.1	13.0
89	Pinocembrin ^a,b,c^	42.89	255.0666	171.0464	151.0040 (80.69), 255.0662 (75.17), 213.0557 (74.89), 145.0662 (70.09), 107.0148 (52.59), 185.0609 (34.69), 169.0660 (24.91), 211.0753 (23.68), 164.0102 (17.93), 187.0757 (16.78), 136.0166 (16.34)	C15H11O4	−0.2	−0.8	10.0
90	Sakuranetin isomer ^b,c^	43.03	285.0769	119.1310	165.1100 (17.55), 150.1056 (7.14), 121.1330 (4.34)	C16H13O5	0.0	−0.1	10.0
91	Pinocembrin chalcone ^b,c^	43.21	255.0668	171.2496	151.0817 (56.63), 107.1704 (32.90), 145.1875 (23.49), 211.2219 (16.51), 169.1781 (15.88), 255.2812 (12.61), 141.1792 (8.45), 213.2497 (8.89), 183.2252 (7.08), 187.1917 (5.71), 133.2347 (3.21)	C15H11O4	−0.5	−1.9	10.0
92	Sakuranetin ^a,b,c^	44.09	285.0773	124.1060	139.1376 (64.17), 145.1010 (42.28), 148.0978 (8.73), 165.1128 (4.71)	C16H13O5	−0.4	−1.6	10.0
93	Galangin ^a,b,c^	44.69	269.0454	269.0454	169.0659 (12.64), 171.0448 (10.87), 213.0554 (10.73), 143.0502 (8.90), 223.0421 (8.03,) 195.0463 (7.34)	C15H9O5	−0.2	−0.8	11.0
94	Genkwanin ^a,b,c^	45.12	283.0619	268.2030	240.1887 (6.18), 117.1122 (5.09), 283.2196 (4.11), 151.0722 (3.32), 148.1045 (1.92)	C16H11O5	−0.7	−2.5	11.0
95	Pinocembrin dihydrochalcone ^b,c^	45.35	257.0826	213.2258	173.1735 (69.29), 171.2012 (39.44), 151.1071 (32.96), 122.1296 (20.68), 156.2112 (21.79), 195.2532 (14.59), 257.2125 (13.28), 239.2599 (12.91), 169.2509 (11.80), 147.2963 (7.77)	C15H13O4	−0.7	−2.6	9.0
96	Caffeic acid pentyl or isopentylester ^b,c^	46.45	249.1138	161.1050	-	C14H17O4	−0.6	−2.3	6.0
97	Unidentified	46.55	269.0449	269.2125	-	C15H9O5	0.6	2.3	11.0
98	Caffeic acid phenethyl ester (CAPE) ^b,c^	46.65	283.0981	135.1231	161.1478 (46.24), 179.1445 (20.40)	C17H15O4	−0.6	−2.0	10.0
99	Pinobanksin 3-O-acetate ^b,c^	47.12	313.0725	253.0510	197.0611 (5.86), 271.0616 (5.36), 209.0610 (4.75), 143.0503 (3.17)	C17H13O6	−0.7	−2.3	16.0
100	Kaempferide (Kaempferol 4′-methyl ether) ^b, c^	47.58	299.0563	284.2046	151.0766 (31.84), 164.0964 (10.53), 107.1859 (6.32), 132.1238 (4.91), 228.1712 (3.34), 299.2162 (3.46), 200.1766 (2.10), 256.1541 (2.02)	C16H11O6	−0.2	−0.7	11.0
101	Methoxychrysin ^b,c^	47.91	283.0614	211.0405	239.0353 (65.55), 268.0380 (8.80)	C16H11O5	−0.2	−0.6	11.0
102	Ermanin (Kaempferol-3,4′-dimethyeter kemferolu) ^b,c^	49.80	313.0719	283.2122	255.1799 (24.32), 253.1653 (17.11), 298.2169 (10.64)	C17H13O6	−0.1	−0.3	11.0
103	*p*-Coumaric acid 3-methyl-3-butenyl ester ^b,c^	50	231.1028	117.1725	119.1277 (90.59), 145.1345 (49.02), 163.1427 (4.99)	C14H15O3	−0.1	−0.4	7.0
104	2-Acetyl-1,3-di-*p*-coumaroylglycerol ^b,c^	50.39	425.1242	163.0403	145.0296 (53.67), 119.0502 (49.02), 219.0658 (11.88), 215.0706 (6.36), 237.0917 (5.21), 171.0817 (5.05), 117.0364 (4.31)	C23H21O8	0.0	0.1	13.0
105	Unidentified	51.10	373.2022	373.4611	137.0954 (28.59), 235.3151 (10.27), 149.1918 (6.47), 93.1141 (4.18)	C22H29O5	−0.2	−0.5	8.0
106	(R/S) 2-Acetyl-3-*p*-coumaroyl-1-feruloylglycerol ^b,c^	51.23	455.1336	163.1189	193.1641 (95.43), 134.1510 (43.39), 119.1319 (41.07), 145.1470 (38.25), 175.3908 (43.52), 160.7224 (15.25)	C24H23O9	1.1	2.5	13.0
107	*p*-Coumaric acid 3-methyl-2-butenyl or 2-methyl-2-butenyl ^b,c^	51.55	231.1027	117.2347	-	C14H15O3	0.0	0.0	7.0
108	(R/S) 1-Acetyl-2,3-di-*p*-coumaroylglycerol ^b,c^	51.67	425.1244	163.1361	145.1342 (64.46), 119.1378 (57.20), 219.2043 (13.02), 171.4749 (7.70)	C23H21O8	−0.2	−0.4	13.0
109	2-Acetyl-1,3-di-feruloylglycerol ^b,c^	52.17	485.1456	193.1733	175.1362 (33.53), 134.1327 (31.79), 149.1651 (12.96), 249.2397 (8.24), 230.3454 (7.88), 160.3150 (7.78), 425.4171 (4.94), 207.1350 (4.01), 470.4230 (4.63)	C25H25O10	−0.3	−0.5	13.0
110	*p*-Coumaric acid benzyl ester ^b,c^	53.31	253.0869	117.2666	145.1076 (12.89), 121.3249 (3.15)	C16H13O3	0.1	0.3	10.0
111	Unidentified	53.75	371.1865	136.0933	92.1420 (23.43)	C22H27O5	−0.1	−0.4	9.0
112	Unidentified	54.50	433.0927	243.2176	271.2540 (40.28), 415.3610 (25.86), 161.1105 (21.48), 253.2210 (10.71), 125.1055 (7.37), 135.1193 (6.47), 165.1139 (5.55), 152.0896 (5.35), 180.0904 (4.98), 227.2045 (4.58), 199.2596 (4.10), 371.2968 (3.52), 280.2369 (2.60)	C24H17O8	0.2	0.4	16.0
113	Caffeic acid cinnamyl ester ^b,c^	55.55	295.0982	134.1352	161.1277 (5.53), 137.1107 (5.18), 106.1119 (4.21)	C18H15O4	−0.6	−1.9	11.0
114	Pinobanksin 3-O-propanoate ^b,c^	57.64	327.0878	253.2179	197.2305 (5.41), 209.2052 (3.72), 271.2717 (2.71), 143.1575 (2.09)	C18H15O6	−0.4	−1.2	11.0
115	*p*-Coumaric acid phenethyl ester ^b,c^	57.93	267.1031	119.1219	145.1261 (81.97), 117.2176 (80.24), 163.1240 (11.83)	C17H15O3	−0.4	−1.6	10.0
116	Caffeic acid hexyl or isohexyl ester ^b,c^	59.36	263.1297	161.1162	135.1220 (75.18)	C15H19O4	−0.8	−3.2	6.0
117	Unidentified	59.43	431.2075	431.4978	153.0866 (95.26), 109.0982 (21.43)	C24H31O7	0.0	0.0	9.0
118	Pinostrobin chalcone ^b,c^	60.13	269.0827	122.0703	165.1175 (83.49), 253.4170 (86.88), 177.1620 (49.29), 226.2073 (47.58), 171.1475 (35.51), 150.0776 (31.31), 163.0634 (21.30), 269.2267 (16.42), 136.1084 (13.47), 198.2301 (14.25)	C16H13O4	−0.3	−0.8	10.0
119	Unidentified	61.72	357.2072	357.4860	243.3735 (78.48), 106.3050 (74.60), 150.1238 (47.95), 175.2599 (45.55), 313.4110 (44.43), 147.2629 (7.94)	C22H29O4	0.0	−0.1	8.0
120	* Flavonoid ^b,c^	61.76	271.0979	152.0937	124.0742 (60.13), 210.2039 (27.77), 238.2594 (25.34), 173.1662 (13.05), 165.1188 (10.13), 271.2509 (7.97), 253.2077 (6.31)	C16H15O4	−0.3	−1.1	9.0
121	*p*-Coumaric acid cinnamyl ester ^b,c^	63.85	279.1029	117.3253	-	C18H15O3	−0.3	−1.0	11.0
122	Pinobanksin 3-O-butanoate or isobutanoate ^b,c^	64.63	341.1037	253.2173	197.2078 (4.89), 209.1812 (3.17)	C19H17O6	−0.6	−1.8	11.0
123	Unidentified	65.17	387.1250	387.4193	267.2677 (77.35), 171.1658 (52.66), 119.1315 (45.75), 283.2991 (40.46), 237.6458 (48.08), 173.1697 (25.91), 197.2909 (27.73), 177.1154 (22.91), 343.3776 (27.15), 293.3467 (19.27), 163.0879 (12.95), 255.2244 (11.74), 145.1501 (10.68)	C24H19O5	−1.2	−3.2	15.0
124	Pinobanksin 3-O-pentenoate or isopentenoate isomer I ^b,c^	65.32	353.1039	253.2231	197.2305 (4.88), 209.1898 (2.96)	C20H17O6	−0.9	−2.5	12.0
125	Pinobanksin 3-O-pentenoate or isopentenoate isomer II ^b,c^	65.65	353.1035	253.2266	271.2152 (26.83), 197.2792 (5.55), 209.5579 (3.51), 225.2615 (2.59)	C20H17O6	−0.5	−1.9	12.0
126	Unidentified	65.86	415.2127	137.1081	415.5088 (82.86), 93.1043 (13.85), 355.4440 (3.98)	C24H31O6	0	−0.1	9.0
127	Unidentified	66.03	445.2235	445.5113	167.1447 (86.18), 151.4185 (63.40), 430.4673 (36.48), 122.4470 (16.79), 108.1295 (5.12), 385.4507 (3.18)	C25H33O7	−0.3	−0.7	9.0
128	Unidentified	66.20	413.1971	161.1344	134.3051 (86.37), 137.2807 (17.10), 179.1723 (16.17), 251.3387 (15.45), 415.3828 (5.43)	C24H29O6	−0.2	−0.4	10.0
129	Unidentified	66.30	373.2026	153.1031	373.4420 (78.55), 109.0952 (32.63), 219.3059 (7.56)	C22H29O5	−0.5	−1.4	8.0
130	Pinobanksin 3-O-benzoate ^b,c^	66.76	375.0878	253.2202	197.1308 (4.84), 225.1950 (3.56), 121.1922 (3.04), 209.1906 (2.85)	C22H15O6	−0.4	−1.0	15.0
131	Pinobanksin derivate ^b,c^	67.45	389.1034	253.2174	271.2537 (50.52)	C23H17O6	−0.3	−0.9	15.0
132	Pinobanksin 3-O-pentanoate or isopentanoate isomer I ^b,c^	67.7	355.1192	253.2167	197.2052 (4.62), 271.2241 (3.55), 209.1801 (2.17)	C20H19O6	−0.5	−1.5	11.0
133	Pinobanksin 3-O-pentanoate or isopentanoate isomer II ^b,c^	67.84	355.1194	253.2180	197.2292 (4.47), 209.1992 (2.52)	C20H19O6	−0.6	−1.8	11.0
134	Unidentified	68.03	315.1606	134.2110	137.0773 (4.72), 179.1280 (2.29)	C19H23O4	−0.4	−1.3	8.0
135	Unidentified	68.11	463.3284	283.4493	-	C24H47O8	−0.8	−1.7	1.0
136	Unidentified	68.35	357.2071	137.1026	357.4751 (89.57), 219.3302 (48.56), 93.1009 (20.22), 149.1908 (10.30), 217.3961 (5.88), 253.1935 (3.44)	C22H29O4	0.0	0.1	8.0
137	Unidentified	68.43	399.2179	339.4431	295.4206 (22.86), 150.1269 (17.15), 357.4646 (8.41), 147.1492 (4.81), 182.1154 (4.93), 107.1210 (4.38), 190.3891 (5.13), 135.2210 (3.67), 189.2491 (3.39), 159.3181 (3.74), 204.2200 (2.91)	C24H31O5	−0.2	−0.6	9.0
138	Pinobanksin 3-O-hexenoate or isohexenoate ^b,c^	68.47	367.1189	253.2181	271.2341 (31.89), 197.2592 (5.77), 209.4797 (3.20), 225.2691 (2.91)	C21H19O6	−0.2	−0.4	12.0
139	Unidentified	68.65	397.2034	145.1304	118.4176 (54.72), 163.1566 (25.26), 251.3224 (17.56), 121.1020 (5.35)	C24H29O5	−1.3	−3.3	10.0
140	Unidentified	68.93	401.1405	119.1448	279.2415 (76.21), 281.4169 (30.38), 254.2454 (22.96), 295.2704 (21.86), 401.4563 (21.60), 267.2166 (10.00), 93.1281 (8.49), 175.3017 (9.94), 297.3369 (8.78), 358.4000 (7.12), 386.3782 (6.84)	C25H21O5	−1	−2.6	15.0
141	Pinobanksin-3-O-hydroxycinnamate ^b,c^	69.14	403.1197	253.2276	271.2222 (4.98), 197.2242 (4.05), 225.3038 (2.92), 149.1545 (2.44)	C24H19O6	−1.0	−2.5	15.0
142	Metoxycinnamic acid cinnamyl ester ^b,c^	69.21	293.2125	293.4701	185.1883 (57.87), 125.1730 (49.45), 141.2221 (18.74), 197.3495 (15.90), 97.2334 (11.61)	C18H29O3	−0.3	−0.9	4.0
143	Unidentified	69.25	471.2384	471.5639					
144	Unidentified	69.51	565.3604	163.1775	119.1081 (27.51), 281.4660 (12.27)	C28H53O11	−1.1	−1.9	2.0
145	Pinobanksin 3-O-hexanoate or isohexanoate isomer I ^b,c^	69.53	369.1347	253.2138	271.2252 (4.95), 197.1623 (3.43), 225.1455 (2.37), 115.1797 (1.95)	C21H21O6	−0.3	−0.8	11.0
146	Unidentified	69.74	471.2396	471.5561	153.1074 (61.63), 109.0816 (11.57), 458.0689 (10.66), 371.3718 (4.17)	C27H35O7	−0.7	−1.5	10.0
147	Pinobanksin 3-O-hexanoate or isohexanoate isomer II ^b,c^	69.82	369.1347	253.2245	197.2037 (4.52), 271.2081 (3.90), 225.2958 (2.22), 209.1639 (1.98), 115.1717 (1.93)	C21H21O6	−0.3	−0.8	11.0
148	Unidentified	70.21	343.2855	283.3972	211.3522 (96.37), 197.2944 (72.36), 253.4190 (30.83), 279.4765 (19.71)	C20H39O4	−0.1	−0.3	1.0
149	Unidentified	70.58	295.2279	295.4866	141.2001 (52.92)	C18H31O3	0.0	−0.1	3.0
150	Pinobanksin 3-O-phenylpentenoate or phenylisopentenoate ester ^b,c^	70.85	429.1344	253.2249	271.2379 (57.79), 197.1788 (3.17), 225.3905 (3.81)	C26H21O6	0.0	−0.1	16.0
151	Unidentified	71.16	455.2449	355.4468	137.1079 (28.97)	C27H35O6	−0.9	−2.1	10
152	Unidentified	71.32	505.3388	283.4757	-	C26H49O9	−0.6	−1.2	2
153	Unidentified	71.37	341.2126	297.4220	341.4479 (55.57), 150.1173 (49.63), 203.4704 (33.41), 107.1309 (27.22), 339.4963 (27.06), 159.4590 (24.45), 183.1562 (14.05), 147.1973 (5.88), 119.1865 (5.82), 189.1923 (4.90)	C22H29O3	−0.4	−1.2	8.0
154	Unidentified	71.81	491.3597	311.5394	-	C26H51O8	−0.8	−1.5	1.0
155	Unidentified	72.15	455.2441	455.5615	137.1079 (82.77), 93.1101 (10.96), 99.1113 (6.76), 355.4591 (5.17)	C27H35O6	−0.2	−0.5	10

**Table legend:** No—number; RT MS—retention tim in uHPLC-MS/MS chromatogram; RBD—ring and double bond equivalents; ^a^ component identified by comparison with standard; ^b^ component identified by comparison with literature; ^c^ component identified by prediction of mass fragment; * component tentatively identified.

**Table 2 molecules-28-02984-t002:** Relatively presence of main propolis components in UHPLC-MS/MS chromatograms of 70EEP from Kazakhstan.

Nb	Component	RT MS	[M-H^+^]^−^	Almaty-1	Almaty-2	Almaty-3	Almaty-4	Almaty-5	Almaty-6	Almaty-7	Bozovoe	Kegen	Kogaly
1	Unidentified	0.88	179.0565	+	tr	+	+	+	+	+	+	+	+
2	Unidentified	1.01	133.0144	tr	tr	-	-	tr	-	-	+	tr	tr
3	Unidentified	1.24	167.0210	-	-	+	tr	-	-	tr	+	-	tr
4	Unidentified	1.43	117.0189	+	-	-	tr	-	-	-	tr	-	tr
5	4-Hydroxybenzoic acid	6.57	137.0246	-	-	-	tr	-	-	-	+	+	+
6	Unidentified	8.01	137.0246	tr	tr	tr	-	tr	tr	-	-	-	-
7	Vanillin isomer	9.33	151.0393	-	-	-	tr	-	-	-	tr	tr	tr
8	Unidentified	9.84	121.0293	-	+	tr	-	tr	-	+	-	-	-
9	Unidentified	10.43	357.1197	-	-	-	tr	-	-	-	-	-	-
10	* Caffeoylglycerol	10.53	253.0717	-	-	-	-	-	-	-	tr	tr	tr
11	Catechin or Epicatechin	10.83	289.0728	-	-	-	+	-	-	-	-	-	-
12	Unidentified	10.87	177.0192	-	-	-	tr	-	-	-	-	tr	tr
13	Unidentified	10.91	165.0557	-	tr	-	-	-	-	-	-	-	-
14	Caffeic acid	11.45	179.0346	+	+	++	++	+	+	+	++	++	++
15	* Caffeoylglycerol	13.03	253.0711	+	tr	tr	tr	+	+	+	+	+	+
16	Unidentified	14.06	195.0663	-	-	-	tr	-	-	-	-	-	-
17	* Caffeic acid dihydroxypentyl or isopentyl ester isomer I	14.33	281.1036	-	-	-	tr	-	-	-	tr	-	-
18	Unidentified	14.35	165.0194	tr	-	-	-	-	-	-	-	-	-
19	*p*-Coumaroylglycerol	14.39	237.0770	-	-	-	-	tr	-	tr	+	-	-
20	*p*-Coumaric acid	14.41	163.0401	-	++	+	tr	++	-	++	++	+	+
21	* Caffeic acid dihydroxypentyl or isopentyl ester isomer II	14.74	281.1034	-	-	-	tr	-	-	-	+	-	tr
22	Ferulic acid	15.19	193.0504	-	+	+	tr	tr	+	+	+	tr	tr
23	* Caffeic acid dihydroxypentyl or isopentyl ester isomer III	15.22	281.1033	tr	-	-	tr	-	tr	-	+	tr	tr
24	Unidentified	15.53	147.0454	-	tr	tr	-	tr	-	+	-	-	-
25	Isoferulic	15.70	193.0503	tr	-	-	-	-	-	-	+	tr	-
26	Unidentified	15.93	205.0509	tr	-	-	-	-	-	-	-	-	-
27	Caffeoylmalic acid (Phaseolic acid) isomer	16.64	295.0827	tr	tr	tr	tr	+	-	+	+	tr	tr
28	Unidentified	17.75	193.0505	-	-	-	tr	-	-	-	-	tr	-
29	Unidentified	18.55	359.1137	-	-	-	-	-	-	tr	-	-	-
30	Eriodyctiol (4′-hydroxynaringenin)	18.82	287.0562	tr	+	tr	tr	+	-	-	+	tr	tr
31	Unidentified	19.63	279.0875	-	+	-	-	-	-	+	-	-	-
32	Apigetrin	21.06	431.0983	-	-	-	-	-	-	-	+	-	-
33	Cinnamic acid	21.21	147.0450	-	tr	-	-	tr	-	-	-	-	-
34	Unidentified	21.37	263.0921	-	-	-	tr	-	-	-	-	-	-
35	Unidentified	21.61	285.0778	-	-	-	-	tr	-	-	+	-	-
36	Unidentified	21.94	349.0931	-	-	tr	-	-	+	-	-	-	+
37	Caffeic acid derivate	22.59	207.0664	-	-	-	+	-	tr	-	-	-	-
38	Unidentified	23.01	287.0560	-	-	-	tr	-	-	-	tr	+	-
39	Pinobanksin 5-methylether	23.32	285.0777	-	++	++	tr	++	-	-	++	++	+
40	Unidentified	23.40	277.1084	-	-	-	tr	-	-	-	-	-	-
41	Unidentified	24.05	277.1088	-	-	-	tr	-	-	-	-	-	-
42	Unidentified	24.56	315.0872	-	tr	tr	-	+	-	-	-	-	-
43	Quercetin	25.05	301.0353	tr	+	tr	tr	+	tr	tr	+	+	tr
44	Luteolin	25.38	285.0412	tr	tr	tr	tr	+	tr	tr	+	+	+
45	Quercetin 3-methyl ether	26.82	315.0497	+	+	+	+	+	+	+	+	+	+
46	Pinobanksin	27.16	271.0615	++	++	++	++	++	++	+++	+++	+++	++
47	Naringenin	28.55	271.0612	+	tr	+	tr	+	tr	+	+	+	+
48	Chrysin-5-methyl-ether	28.75	267.0662	-	-	-	-	-	-	-	+	tr	-
49	Unidentified	28.76	301.0721	-	+	+	-	+	-	-	tr	-	-
50	1-Caffeoyl-3-p-coumaroylglycerol	28.89	399.1085	-	+	-	-	-	-	tr	-	-	-
51	Unidentified	29.48	269.0822	-	+	tr	-	+	-	-	+	+	-
52	Caffeic acid propyl or isopropyl ester	29.80	221.0826	-	-	tr	-	tr	-	+	-	-	-
53	Unidentified	30.21	301.0717	-	-	tr	-	tr	-	+	tr	tr	-
54	Unidentified	30.39	389.1977	+	-	-	-	-	tr	-	-	-	-
55	Apigenin	30.47	269.0457	+	+	+	+	+	+	+	++	+	+
56	Kaempferol	31.14	285.0405	+	+	+	+	+	+	+	++	+	+
57	Isorhamnetin	31.72	315.0509	-	+	+	tr	+	-	-	+	+	tr
58	Unidentified	31.86	387.1814	+	-	-	-	-	+	-	-	-	+
59	Quercetin-methyl-ether	32.23	315.0511	-	tr	+	tr	+	-	+	+	+	-
60	Unidentified	32.82	417.1928	+	-	-	-	-	tr	-	-	-	-
61	Luteolin-5-methyl ether	32.97	299.0549	+	+	+	+	+	+	tr	++	+	+
62	(R/S) 1,2-di-p-Coumaroylglycerol isomer I	33.02	383.1137	-	-	-	-	-	-	tr	-	-	-
63	Caffeic acid buten or isobuten ester	33.60	233.0830	-	tr	tr	-	tr	-	tr	-	-	-
64	Quercetin-di-methyl-ether	33.68	329.0669	-	+	+	+	+	tr	-	+	+	tr
65	1,3-di-p-Coumaroylglycerol	33.91	383.1143	-	+	tr	-	tr	-	++	tr	-	-
66	Unidentified	33.93	373.2015	+	-	-	-	-	+	-	-	tr	tr
67	Unidentified	33.98	359.0777	-	-	-	tr	-	-	-	-	-	-
68	Unidentified	34.35	391.2134	tr	-	-	-	-	tr	-	-	-	-
69	Galangin-5-methyl-ether	34.37	283.0612	-	+	tr	-	+	-	-	+	+	tr
70	(R/S) 1-p-Coumaroyl-3-feruloylglycerol	34.38	413.1241	-	tr	-	-	-	-	+	-	-	-
71	(R/S) 1,2-di-p-Coumaroylglycerol isomer II	34.39	383.1137	-	+	-	-	-	-	tr	-	-	-
72	5-Methyl-pinobanksin-3- acetate	34.61	327.0878	-	tr	tr	-	tr	-	-	+	+	-
73	2-Acetyl-1,3-di-caffeoylglycerol	35.15	457.1141	-	+	tr	-	tr	-	+	-	-	-
74	Unidentified	36.21	229.0874	-	-	+	-	-	+	-	-	-	+
75	Rhamnetin	36.48	315.0509	+	tr	tr	tr	tr	tr	-	+	+	tr
76	Kaempferol-methyl-ether	36.64	299.0563	+	-	-	tr	-	-	-	+	+	tr
77	Caffeic acid butyl or isobutyl ester isomer isomer I	37.17	235.0976	-	-	tr	tr	-	-	tr	tr	tr	-
78	Caffeic acid prenyl derivate	37.70	247.0975	tr	-	-	+	-	tr	-	+	+	+
79	Caffeic acid butyl or isobutyl ester isomer isomer II	37.92	235.0978	-	+	tr	-	+	-	+	-	-	-
80	Unidentified	38.17	373.2012	+	-	-	-	-	+	-	-	tr	tr
81	Quercetin-dimethyl-ether	38.75	329.0669	tr	+	tr	+	tr	tr	tr	+	+	tr
82	Caffeic acid 2-methyl-2-butenyl ester	39.04	247.0979	++	++	++	+++	++	++	++	+++	+++	+++
83	Caffeic acid 3-methyl-2-butenyl ester (Basic prenyl ester)	40.44	247.0979	++	++	++	+++	++	++	++	+++	+++	+++
84	Caffeic acid 3-methyl-3-butenyl ester	40.90	247.0977	+	+	+	+	+	+	+	++	++	+
85	(R/S) 2-Acetyl-1-caffeoyl-3-p-coumaroylglycerol	41.69	441.1197	-	+	tr	-	-	-	+	-	-	-
86	Chrysin	42.00	253.0505	+	++	++	+	++	+	++	+++	+++	+
87	Caffeic acid benzyl ester	42.28	269.0818	tr	tr	tr	+	tr	tr	tr	+	+	+
88	(R/S) 2-Acetyl-1-caffeoyl-3-feruloylglycerol	42.49	471.1297	-	+	-	-	-	-	tr	-	-	-
89	Pinocembrin	42.89	255.0666	++	++	++	++	+++	++	+++	+++	+++	+
90	Sakuranetin isomer	43.03	285.0769	-	-	-	-	tr	-	-	-	-	-
91	Pinocembrin chalcone	43.21	255.0668	-	-	++	-	-	-	-	-	-	-
92	Sakuranetin	44.09	285.0773	tr	++	++	tr	++	tr	+	+	+	tr
93	Galangin	44.69	269.0454	++	++	++	++	++	+	++	+++	++	+
94	Genkwanin	45.12	283.0619	tr	tr	tr	-	+	-	tr	+	tr	tr
95	Pinocembrin dihydrochalcone	45.35	257.0826	-	tr	+	-	tr	-	-	tr	-	-
96	Caffeic acid pentyl or isopentylester ester	46.45	249.1138	tr	tr	tr	tr	tr	tr	+	+	+	tr
97	Unidentified	46.55	269.0449	-	-	tr	tr	tr	-	-	tr	+	tr
98	Caffeic acid phenethyl ester (CAPE)	46.65	283.0981	+	-	-	++	tr	+	-	++	++	++
99	Pinobanksin 3-O-acetate	47.12	313.0725	++	+++	+++	+	+++	++	+++	+++	+++	+++
100	Kaempferide (Kaempferol 4′-methyl ether)	47.58	299.0561	-	tr	tr	-	+	-	tr	tr	-	-
101	Methoxychrysin	47.91	283.0614	-	tr	tr	+	tr	tr	tr	+	+	tr
102	Ermanin (Kaempferol-3,4′-dimethyeter kemferolu)	49.80	313.0719	tr	-	-	tr	-	-	-	+	tr	tr
103	*p*-Coumaric acid 3-methyl-3-butenyl ester	50.00	231.1028	-	-	-	+	-	-	-	+	tr	tr
104	2-Acetyl-1,3-di-p-coumaroylglycerol	50.39	425.1242	-	++	+	-	+	-	+++	+	-	-
105	Unidentified	51.10	373.2022	++	-	-	-	-	+	-	-	+	tr
106	(R/S) 2-Acetyl-3-p-coumaroyl-1-feruloylglycerol	51.23	455.1336	-	+	+	-	tr	-	+	-	-	-
107	p-Coumaric acid 3-methyl-2-butenyl or 2-methyl-2-butenyl ester	51.55	231.1027	-	-	-	+	-	-	-	+	+	tr
108	(R/S) 1-Acetyl-2,3-di-p-coumaroylglycerol	51.67	425.1244	-	tr	-	-	-	-	+	-	-	-
109	2-Acetyl-1,3-di-feruloylglycerol	52.17	485.1456	-	tr	-	-	-	-	tr	-	-	-
110	*p*-Coumaric acid benzyl ester	53.31	253.0869	-	++	++	-	++	-	+++	+	tr	tr
111	Unidentified	53.75	371.1865	+	-	-	-	-	+	-	-	tr	tr
112	Unidentified	54.50	433.0927	+	-	-	+	-	tr	-	+	+	tr
113	Caffeic acid cinnamyl ester	55.55	295.0982	-	tr	+	+	+	-	-	+	+	+
114	Pinobanksin 3-O-propanoate	57.64	327.0878	+	+	++	+	++	+	+	++	++	+
115	*p*-Coumaric acid phenethyl ester	57.93	267.1031	-	++	+	-	++	-	+	+	-	tr
116	Caffeic acid hexyl or isohexyl ester	59.36	263.1297	-	tr	tr	-	tr	-	tr	-	-	-
117	Unidentified	59.43	431.2075	+	-	-	-	-	+	-	-	tr	tr
118	Pinostrobin chalcone	60.13	269.0827	-	+	+++	tr	++	-	-	-	-	-
119	Unidentified	61.72	357.2072	+	-	-	-	-	tr	-	-	-	-
120	* Flavonoid	61.76	271.0979	-	+	++	-	++	-	-	+	-	-
121	*p*-Coumaric acid cinnamyl ester	63.85	279.1029	-	+++	+++	-	+++	-	-	++	+	tr
122	Pinobanksin 3-O-butanoate or isobutanoate	64.63	341.1037	+	+	++	++	++	+	++	++	++	+
123	Unidentified	65.17	387.1250	-	+	+	-	+	-	-	tr	-	-
124	Pinobanksin 3-O-pentenoate or isopentenoate isomer I	65.32	353.1039	-	+	+	+	+	tr	-	+	+	+
125	Pinobanksin 3-O-pentenoate or isopentenoate isomer II	65.65	353.1035	-	-	-	-	-	-	-	+	+	-
126	Unidentified	65.86	415.2127	++	-	-	-	-	+	-	-	+	+
127	Unidentified	66.03	445.2235	+	-	-	-	+	+	-	-	tr	tr
128	Unidentified	66.20	413.1971	-	-	-	tr	-	-	-	+	+	-
129	Unidentified	66.30	373.2026	+	-	-	-	-	+	-	-	+	tr
130	Pinobanksin 3-O-benzoat	66.76	375.0878	-	+	tr	-	+	-	-	+	tr	-
131	Pinobanksin derivate	67.45	389.1034	-	-	-	-	+	-	-	+	-	-
132	Pinobanksin 3-O-pentanoate or isopentanoate isomer I	67.70	355.1192	+	+	+	+	+	+	++	+	+	+
133	Pinobanksin 3-O-pentanoate or isopentanoate isomer II	67.84	355.1194	+	+	+++	++	++	+	+	++	+++	++
134	Unidentified	68.03	315.1606	+	-	-	+	-	+	-	+	+	+
135	Unidentified	68.11	463.3284	-	-	-	-	+	-	+	+	+	+
136	Unidentified	68.35	357.2071	+	-	-	-	-	+	-	-	-	-
137	Unidentified	68.43	399.2179	+	-	-	-	-	+	-	-	-	-
138	Pinobanksin 3-O-hexenoate or isohexenoate	68.47	367.1189	-	-	-	-	-	-	-	+	-	-
139	Unidentified	68.65	397.2034	-	+	++	-	+	-	-	-	-	-
140	Unidentified	68.93	401.1405	-	+	++	-	+	-	-	-	-	-
141	Pinobanksin-3-O-hydroxycinnamate	69.14	403.1197	-	tr	tr	+	tr	tr	-	+	+	+
142	Metoxycinnamic acid cinnamyl ester	69.21	293.2125	tr	+	++	tr	+	tr	-	++	+	+
143	Unidentified	69.51	565.3604	-	-	-	++	-	-	-	-	-	-
144	Pinobanksin 3-O-hexanoate or isohexanoate isomer I	69.53	369.1347	-	-	-	-	tr	-	-	+	+	tr
145	Unidentified	69.74	357.2082	+++	-	-	+	+	+++	-	tr	+++	+++
146	Pinobanksin 3-O-hexanoate or isohexanoate isomer II	69.82	369.1347	-	+	++	-	+	-	+	+	tr	tr
147	Unidentified	70.03	477.3439	-	-	-	+	-	-	+	-	-	-
148	Unidentified	70.21	343.2855	-	tr	+	+	+	-	+	+	+	+
149	Unidentified	70.58	295.2279	-	+	tr	-	tr	-	-	+	+	tr
150	Pinobanksin 3-O-phenylpentenoate or phenylisopentenoate ester	70.85	429.1344	-	-	-	-	-	-	-	+	tr	+
151	Unidentified	71.16	455.2449	+	-	-	-	-	+	-	-	tr	-
152	Unidentified	71.32	505.3388	-	+	++	+	-	-	+	-	-	-
153	Unidentified	71.37	341.2126	+++	-	-	tr	-	+++	-	tr	+	++
154	Unidentified	71.81	491.3597	-	-	-	+	+	-	+	-	-	-
155	Unidentified	72.15	455.2441	++	-	-	-	-	++	-	-	+	+

**Table legend**: RT MS—retention tim in uHPLC-MS/MS chromatogram; - component absent; tr component present in traces; + component present in low amount; ++ component present in average amount; +++ component present in high amount.

**Table 3 molecules-28-02984-t003:** Matrix of main components in UHPLC-DAD profile of Kazakh propolis 70% ethanol in water extract in 280 nm *.

Component	Almaty-1	Almaty-2	Almaty-3	Almaty-4	Almaty-5	Almaty-6	Almaty-7	Bozovoe	Kegen	Kogaly
Caffeic acid	1.63	0.74	1.13	8.67	0.91	3.77	1.62	8.09	10.19	8.44
*p*-Coumaric acid	0.00	11.63	8.28	0.24	10.57	0.18	16.43	0.62	0.73	0.70
Benzoic acid ^NI^	0.00	0.41	0.14	0.00	0.17	0.00	1.11	0.00	0.00	0.00
Ferulic acid	0.00	2.39	0.74	0.74	0.52	1.58	4.74	0.33	0.41	0.59
Isoferulic	1.91	0.00	0.00	0.43	0.10	0.23	0.00	0.49	0.55	0.50
Caffeic acid ethyl ester ^NI^	0.71	0.00	0.00	0.83	0.00	0.13	0.00	0.77	1.08	0.54
Cinnamic acid	0.14	24.70	13.12	0.07	17.23	0.00	0.48	0.31	0.73	0.53
Pinobanskin-5-methyl-ether	0.00	1.32	1.37	0.13	1.81	0.00	0.00	0.99	0.50	0.51
Unidentified, UV = 300 nm	0.00	0.00	0.00	0.00	0.00	4.23	0.00	0.00	0.41	3.69
Pinobanksin	2.57	0.89	1.21	2.97	1.57	3.42	2.60	5.55	3.09	4.01
1,3-di-*p*-Coumaroylglycerol	0.21	0.28	0.15	0.00	0.00	0.00	1.83	0.00	0.00	0.00
Unidentifeid, UV = 330 nm	0.00	0.00	0.16	0.00	0.00	4.02	0.00	0.00	0.25	2.16
Caffeic acid 2-methyl-2-butenyl ester	5.71	0.34	1.11	16.63	0.09	7.31	1.04	4.68	13.39	9.96
Caffeic acid 3-methyl-2-butenyl ester	8.62	0.85	1.63	23.43	1.34	10.10	1.86	6.46	18.36	14.29
Caffeic acid 3-methyl-3-butenyl ester	1.45	0.29	0.60	2.38	0.45	0.91	0.54	0.97	1.43	0.81
Chrysin	5.83	2.55	4.01	1.96	4.44	2.35	3.67	10.76	3.63	3.15
Caffeic acid benzyl ester	0.19	0.00	0.00	0.65	0.00	0.51	0.48	1.11	0.68	0.24
Pinocembrin	6.99	3.38	5.31	6.45	5.91	5.67	5.05	13.56	6.89	5.44
Pinocembrin chalcone	0.00	0.00	1.55	0.00	0.17	0.00	0.00	0.00	0.00	0.00
Sakuranetin	0.00	2.10	2.67	0.21	2.96	0.94	0.49	1.22	0.45	0.60
Galangin	2.34	2.24	4.27	3.80	3.58	2.16	2.52	4.45	3.53	2.86
Unidentified flavonoid, UV = 272 nm, 328 nm	0.00	0.00	0.00	0.00	0.00	0.00	1.18	0.00	0.00	0.00
Caffeic acid phenethyl ester (CAPE)	1.87	0.00	0.37	2.62	0.18	1.31	0.00	1.22	2.03	2.09
Pinobanksin 3-O-acetate	8.67	3.89	8.73	14.36	6.80	8.96	7.25	14.49	12.98	9.90
Kaempferide (Kaempferol 4′-methyl ether)	0.00	0.78	0.96	0.00	1.18	0.00	0.00	0.00	0.00	0.00
2-Acetyl-1,3-di-*p*-coumaroylglycerol	0.00	2.69	0.79	0.00	0.31	0.00	7.16	0.20	0.00	0.00
Unidentified, UV = 285 nm	2.42	0.00	0.00	0.00	0.00	0.00	0.00	0.00	0.00	0.00
(R/S) 2-Acetyl-3-*p*-coumaroyl-1-feruloylglycerol	0.00	1.35	0.00	0.00	0.00	0.00	2.95	0.00	0.00	0.00
Unidentified, UV = 324 nm	0.00	0.00	1.40	0.00	0.00	8.67	0.00	0.00	1.03	8.09
*p*-Coumaric acid benzyl ester	0.00	2.40	1.46	0.00	1.64	0.00	17.07	0.00	0.00	0.00
Ferulic acid benzyl ester	0.00	0.26	0.40	0.34	0.21	0.00	2.91	0.35	0.00	0.00
Pinobanksin-3-O-propanoate	0.00	0.23	1.06	0.47	0.98	0.16	0.52	1.09	0.73	0.48
*p*-Coumaric acid phenethyl ester	0.00	1.56	1.00	0.03	1.15	0.00	0.00	0.07	0.00	0.00
Pinostrobin chalcone	0.00	0.08	1.46	0.15	0.39	0.00	0.00	0.00	0.00	0.00
Tectochrysin ^NI^	0.74	0.00	0.07	0.43	0.17	0.35	0.36	2.06	0.74	0.66
Pinostrobin ^NI^	0.43	1.13	2.68	0.86	3.07	0.41	5.14	0.99	0.56	0.53
*p*-Coumaric acid cinnamyl ester	0.00	8.37	7.84	0.00	9.84	0.00	0.00	0.16	0.39	0.33
Pinobanksin 3-O-butanoate or isobutanoate	0.30	0.46	1.30	0.45	0.92	0.35	0.94	1.00	0.44	0.39
Unidentified, UV = 258 nm	1.13	0.00	0.00	0.00	0.00	0.42	0.00	0.00	0.00	0.00
Unidentified, UV = 262, 293 nm	2.29	0.38	0.72	0.00	0.73	0.88	0.00	0.00	0.00	0.00
Unidentified, UV = 258 nm	1.84	0.00	0.00	0.00	0.00	0.84	0.00	0.00	0.19	0.25
Unidentified, UV = 279 nm	0.00	3.76	0.00	0.00	0.00	0.00	1.34	0.00	0.00	0.00
Pinobanksin 3-O-pentanoate or isopentenoate II	0.30	0.78	0.71	0.11	1.15	1.17	1.28	2.12	1.60	1.08
Unidentified, UV = 310 nm	0.00	1.14	1.33	0.00	1.64	0.00	0.00	0.00	0.00	0.00
*p*-Methoxy cinnamic acid cinnamyl ester	0.30	1.91	2.80	0.33	2.50	0.20	0.40	1.55	0.65	0.60
Unidentified, UV = 257 nm	17.60	0.00	0.00	0.00	0.00	11.68	0.00	2.39	1.85	2.89
Unidentified, UV = 282 nm	0.00	0.00	1.54	0.00	0.00	0.00	0.00	0.00	0.00	0.00
Unidentified, UV = 262 nm	3.59	0.00	0.00	0.00	0.00	0.00	0.00	0.00	0.27	0.45
Unidentified, UV = 267 nm	0.00	0.00	0.00	0.00	2.55	0.00	0.00	0.00	0.00	0.00
Unidentified, UV = 240 nm	0.00	2.81	1.50	0.00	0.00	0.00	0.00	0.00	0.00	0.00
Unidentified, UV = 258 nm	4.71	0.00	0.00	0.00	0.00	3.15	0.00	0.24	0.41	0.70
Unidentified, UV = 257 nm	1.06	0.00	0.00	0.00	0.00	0.00	0.00	0.00	0.09	0.22

**Table legend:** * component presence was described as % of UV chromatograms (280 nm) relatively peak area; ^NI^—component did not produce ions in negative mode in experimental conditions (it was not presented in Table 1 and Table 2).

**Table 4 molecules-28-02984-t004:** Antimicrobial activity of Kazakh propolis *.

	Cluster 1	Cluster 2	Cluster 3
Almaty-1	Almaty-4	Kegen	Kogaly	Almaty-6	Bozove	Almaty-2	Almaty-3	Almaty-5	Almaty-7
Gram-Positive Bacteria	MIC	MBC/MFC	MIC	MBC/MFC	MIC	MBC/MFC	MIC	MBC/MFC	MIC	MBC/MFC	MIC	MBC/MFC	MIC	MBC/MFC	MIC	MBC/MFC	MIC	MBC/MFC	MIC	MBC/MFC
*S. aureus* ATCC 25923	62.5	62.5	62.5	62.5	62.5	125	>4000	ND	62.5	62.5	62.5	62.5	125	250	62.5	62.5	62.5	62.5	62.5	62.5
*S. aureus* ATCC 43300	62.5	62.5	62.5	62.5	125	125	>4000	ND	62.5	62.5	31.3	62.5	125	250	62.5	62.5	62.5	125	62.5	62.5
*S. epidermidis* ATCC 12228	62.5	62.5	62.5	125	125	125	>4000	ND	62.5	125	62.5	125	125	250	62.5	62.5	62.5	125	62.5	125
*M. luteus* ATCC 10240	31.3	31.3	31.3	62.5	31.3	62.5	>4000	ND	31.3	62.5	31.3	62.5	62.5	2000	31.3	31.3	31.3	4000	31.3	62.5
*B. cereus* ATCC 10876	62.5	2000	62.5	4000	125	4000	>4000	ND	31.3	1000	31.3	125	62.5	>4000	31.3	>4000	31.3	2000	31.3	1000
*E. faecalis* ATCC 29212	62.5	>4000	125	250	125	500	>4000	ND	125	>4000	62.5	125	250	>4000	125	250	125	250	125	>4000
**Gram-negative bacteria**																				
*S. typhimurium* ATCC 14028	>4000	ND	>4000	ND	>4000	ND	>4000	ND	4000	ND	>4000	ND	>4000	ND	4000	ND	>4000	ND	4000	ND
*E. coli* ATCC 25922	>4000	ND	4000	ND	>4000	ND	>4000	ND	4000	ND	>4000	ND	>4000	ND	>4000	ND	>4000	ND	4000	ND
*P. mirabilis* ATCC 12453	>4000	ND	>4000	ND	>4000	ND	>4000	ND	>4000	ND	>4000	ND	4000	ND	>4000	ND	4000	ND	>4000	ND
*K. pneumoniae* ATCC 13883	>4000	ND	>4000	ND	>4000	ND	>4000	ND	4000	ND	4000	ND	>4000	ND	>4000	ND	>4000	ND	4000	ND
*P. aeruginosa* ATCC 9027	>4000	ND	4000	ND	>4000	ND	>4000	ND	>4000	ND	>4000	ND	4000	ND	>4000	ND	4000	ND	>4000	ND
*H.pylori* ATCC 43504	62.5	62.5	62.5	62.5	62.5	63.5	31.3	31.3	31.3	31.3	31.3	31.3	31.3	31.3	31.3	31.3	31.3	31.3	62.5	62.5
**Yeasts**																				
*C. glabrata* ATCC 90030	500	1000	125	250	500	250	125	250	250	500	125	125	500	1000	250	250	500	2000	250	500
*C. albicans* ATCC 102231	1000	2000	125	250	1000	500	250	500	500	1000	125	250	500	1000	500	1000	1000	1000	500	1000
*C. parapsilosis* ATCC 22019	500	2000	125	125	250	1000	125	250	250	1000	62.5	250	500	4000	1000	1000	250	1000	250	1000

**Table legend:** * values of MIC, MBC and MFC were expressed as µg/mL.

**Table 5 molecules-28-02984-t005:** Antibacterial activity (MIC) against reference *H. pylori* strain and inhibition of *H. pylori* urease activity (IC_50_) by tested 70EEP.

	Propolis Sample	IC_50_ (µg/mL)	MIC (µg/mL)
**Cluster 1**	Almaty-1	440.73	62.5
Almaty-4	509.92	62.5
Kegen	918.46	32.5
Kogaly	525.50	31.3
Almaty-6	11,177.24	31.3
Bozove	739.31	31.3
**Cluster 2**	Almaty-2	6030.86	31.3
Almaty-3	8465.38	31.3
Almaty-5	743.51	31.3
**Cluster 3**	Almaty-7	4089.94	62.5
Thiourea	92.7	--

## Data Availability

The data presented in this study are available on request from the authors.

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
