# Peer review of "Phytochemical Profile and Antimicrobial Potential of Propolis Samples from Kazakhstan"

_molecules, 2023, doi:10.3390/molecules28072984_

Round 1

Reviewer 1 Report

This experimental study investigates the phytochemical profile and antimicrobial properties of 11 propolis samples collected in Kazakhstan.

Different types of propolis from various regions/continents are introduced (e.g., poplar type, aspen type, birch, green, red type, etc.), emphasizing the potential use in medicinal purposes due to the many biological activities of these nutraceuticals.

In the Results and Discussion section, Tables are not listed well and do not correspond with the main text! Some components are missing in Table 1 (e.g., No 21 and No for Caffeic acid cinnamyl ester). Table 3 (Presence of main components in UHPLC-DAD profile of Kazakh propolis 70% ethanol in water extract) is mentioned, but it doesn’t exist. The authors should pay attention to technical elements.

Antimicrobial activity of propolis samples is nicely described and compared to previous literature data.

The methodology provides necessary data about collecting row material, preparation of propolis extracts, assays and analysis. It remains unclear how many samples you have analyzed, 10 or 11? I can not notice Almaty-2 in your results. Please clarify this!

Moderate editing of English language and style required.

The paper complies with the field of the journal, but this form is not ready for publication.

Author Response

Reviewer 1

This experimental study investigates the phytochemical profile and antimicrobial properties of 11 propolis samples collected in Kazakhstan.

Different types of propolis from various regions/continents are introduced (e.g., poplar type, aspen type, birch, green, red type, etc.), emphasizing the potential use in medicinal purposes due to the many biological activities of these nutraceuticals.

In the Results and Discussion section, Tables are not listed well and do not correspond with the main text! Some components are missing in Table 1 (e.g., No 21 and No for Caffeic acid cinnamyl ester). Table 3 (Presence of main components in UHPLC-DAD profile of Kazakh propolis 70% ethanol in water extract) is mentioned, but it doesn’t exist. The authors should pay attention to technical elements.

A: Thank you for the issues. Table 3 disappeared during the edition. It was added. Minor errors were also corrected.

Antimicrobial activity of propolis samples is nicely described and compared to previous literature data.

The methodology provides necessary data about collecting row material, preparation of propolis extracts, assays, and analysis. It remains unclear how many samples you have analyzed, 10 or 11? I can not notice Almaty-2 in your results. Please clarify this!

A: Thank you for the issues. We investigated 10 samples. Their numbering was corrected in the whole manuscript.

Moderate editing of English language and style required.

A: Thank You for the suggestion. The English language was corrected by a native speaker.

The paper complies with the field of the journal, but this form is not ready for publication.

A: Thank You for the suggestion. The form of the manuscript has been corrected according to reviewers suggestions.

Reviewer 2 Report

The article “Phytochemical profile and antimicrobial potential of propolis samples from Kazakhstan” by JarosÅ‚aw Widelski at al. is of good quality and clear. I recommend this paper to be published in the journal. Here are some suggestions:

1: What motivated the authors to prepare the article. More recent research information and references on advantages and uniqueness of Kazakh propolis should be added in “Introduction” to highlight the novelty of this work clearly.

2: More detailed experimental data or spectrograms should be provided. For example, the IC50 values of the main active ingredient of Kazakh propolis should be added.

3: The “Conclusions” of the manuscript is weak and should be improved.

Author Response

Reviewer 2

The article “Phytochemical profile and antimicrobial potential of propolis samples from Kazakhstan” by JarosÅ‚aw Widelski at al. is of good quality and clear. I recommend this paper to be published in the journal. Here are some suggestions:

1: What motivated the authors to prepare the article. More recent research information and references on the advantages and uniqueness of Kazakh propolis should be added in the “Introduction” to highlight the novelty of this work clearly.

A: Thank you for the valuable issue. Our main motivation was a very small amount of data concerning propolis from this region.

2: More detailed experimental data or spectrograms should be provided. For example, the IC50 values of the main active ingredient of Kazakh propolis should be added.

A: Thank you for the valuable issue. We want to investigate IC50 and another property of singular components of propolis in the future. In the current manuscript, our main purpose was a screening of natural extracts and searching potential components for further investigations. We add some chromatograms of representative samples.

3: The “Conclusions” of the manuscript are weak and should be improved.

A: Thank you for the valuable issue. We improved this section due to your suggestions.

Reviewer 3 Report

We congratulate the authors for the study and for choosing this research topic, of major interest major for medical-pharmaceutical field.

We agree with the publication of the manuscript in Molecules Journal, only after the following revisions as follows:

- completing the Conclusions section so that any reader of the article understands the purpose of this study and what perspectives are opened for the practical applicability of the sorts studied, based on the results obtained, from the perspective of evidence-based traditional medicine.

- reconsider Tables 1 and 2 in a shorter form

- line 7 – remove space before comma

- 16 – remove underline

- 14-16, 18, 23 – hyperlink for email addresses

- 26 – correct form – paper

- (rule applied for the entire manuscript) describe any acronym at their very first entry, for Abstract and main text

- (rule applied for the entire manuscript) – anti-Helicobacter

- m4 – Aspen poplar (Populus tremula); general rule – Aspen (with capital letter)

- 43 – remove the last comma

- (rule applied for the entire manuscript) – there is no plural for propolis – use propolis types or find another form

- 50 – saliva

- 55 – American English – color, odor

- 56 – tus ?

- 58 – time of collection – harvest time

- 63 – correct form – Zealand; Birtch – correct form birch (?)

- 66 – correct form – e.g.,

- 77 – (rule applied for the entire manuscript) remove space before comma

- 79 – cancer [9] (comma)

- 81 – "Propolis in different preparations are used" – subject-verb agreement (rule applied for the entire manuscript)

92 – correct form – Kazakhstan

100, 104 – use comma after "Generally"

105 – correct form – ionization

127-128 – correct form "Chemical composition of analyzed propolis types"

131 – correct form – caffeic acid

139 – correct form – "occurrence"

143 – e.g.,

148 – correct forms – "Comparative analysis of chemical composition of extracts for Kazakh propolis samples"

- Figure 1 – explanations for Dlink, Dmax

159 – height

Table 1 – use subscript

Table 2 – correct form – Luteolin

214 – correct form – antibacterial

228-236 – correct forms – comparison, Interestingly, compounds, extracts, characterized, compounds

260 – S. aureus (space between)

261 – correct form - Another study confirmed / Other studies confirmed

263 – correct form – values

270 – correct form - These findings confirmed / This finding confirmed

379 – correct form - inhibitory

381 – correct form - Kazakhstan is distinctly

412 – correct form – extraction obtained

423 - solution of sodium formate (?)

488-493 – analysed, bacteriai (?)

458 – correct form 5 mL

- for References – complete the entire section of authors (without suspension)< also, remove space after surnames, remove DOI (see https://www.mdpi.com/journal/molecules/instructions)< remove any underlined section

Author Response

Reviewer 3
We congratulate the authors for the study and for choosing this research topic, of major interest major for medical-pharmaceutical field.

A: Thank You for such nice words.

We agree with the publication of the manuscript in Molecules Journal, only after the following revisions as follows:

- completing the Conclusions section so that any reader of the article understands the purpose of this study and what perspectives are opened for the practical applicability of the sorts studied, based on the results obtained, from the perspective of evidence-based traditional medicine.

A: Thank You for your suggestion. We corrected “Conlusions”. Section.

- reconsider Tables 1 and 2 in a shorter form

- line 7 remove space before comma

- 16 remove underline

- 14-16, 18, 23 hyperlink for email addresses

- 26 correct form paper

- (rule applied for the entire manuscript) describe any acronym at their very first entry,
for Abstract and main text

- (rule applied for the entire manuscript) anti-Helicobacter

- m4 Aspen poplar (Populus tremula); general rule Aspen (with capital letter)

- 43 remove the last comma

- (rule applied for the entire manuscript) there is no plural for propolis use propolis types or
find another form

- 50 saliva

- 55 American English color, odor

- 56 tus ?

- 58 time of collection harvest time

- 63 correct form Zealand; Birtch correct form birch (?)

- 66 correct form e.g.,

- 77 (rule applied for the entire manuscript) remove space before comma

- 79 cancer [9] (comma)

- 81 "Propolis in different preparations are used" subject-verb agreement (rule applied for
the entire manuscript)

92 correct form Kazakhstan

100, 104 use comma after "Generally"

105 correct form ionization

127-128 correct form "Chemical composition of analyzed propolis types"

131 correct form caffeic acid

139 correct form "occurrence"

143 e.g.,

148 correct forms "Comparative analysis of chemical composition of extracts for Kazakh
propolis samples"

- Figure 1 explanations for Dlink, Dmax

159 height

Table 1 use subscript

Table 2 correct form Luteolin

214 correct form antibacterial

228-236 correct forms comparison, Interestingly, compounds, extracts, characterized,
compounds

260 S. aureus (space between)

261 correct form - Another study confirmed / Other studies confirmed

263 correct form values

270 correct form - These findings confirmed / This finding confirmed

379 correct form - inhibitory

381 correct form - Kazakhstan is distinctly

412 correct form extraction obtained

423 - solution of sodium formate (?)

488-493 analysed, bacteriai (?)

458 correct form 5 mL

A: Thank you for your valuable comments and insightful reading of the manuscript. We have made the
suggested corrections.

- for References complete the entire section of authors (without suspension)< also, remove
space after surnames, remove DOI
(see https://www.mdpi.com/journal/molecules/instructions)< remove any underlined section

A: Thank you for your valuable comments. Corrections have been made

Reviewer 4 Report

1. The English need improvement since there are some grammatical and syntax errors in the manuscript. For example, in line number 26, the word “current” may be as “the current”; in line number 29, “serial” as “the serial”; in line number 30, “strong” as “a strong”; in line number 31, “for poplar” as “of poplar”; in line number 33, “Second” as “The second”; in line number 40, “medicament” as “a medicament”; in line number 62, “the Central” as “Central”; in line number 73, “3000” as “in 3000”; in line number 74, “in the” as “the”; in line number 75, “ingredient of” as “an ingredient in”; in line number 76, “according” as “according to”; in line number 77, “treatment” as “the treatment”; in line number 84, “bioactivities” as “the bioactivities”; in line number 84, “is related” as “are related”; in line number 84, “presence” as “the presence”; in line number 86, “antibacterial effect” as “an antibacterial effects”; in line number 100, “substances” as “the substances”; in line number 104, “components” as “the components”; in line number 105, “very” as “a very”; in line number 200, “same” as “the same”; in line number 107 and 108,  “current paper” as “the current paper,”; in line number 108, “main” as “the main”; in line number 109, “high” as “the high”; in line number 110, “trace” as “a trace”; in line number 114, “hydroxycinnamic” as “the hydroxycinnamic”; in line number 119, “flavonoid” as “the flavonoid”; in line number 119, “components” as “the components”; in line number 121, “Rest” as “The rest”; in line number 121, “of common” as “of the common”; in line number 124, “Same” as “The same”; in line number 125, “rest” as “the rest”; in line number 126, “such as” as “as”; in line number 127, “Chemical” as “The chemical”; in line number 128, “main” as “the main”; in line number 133, “Apart of” as “Apart from”; in line number 139, “presence” as “the presence”; in line number 135, “strong” as “a strong”; in line number 139, “high” as “a high”; in line number 140, “plant” as “the plant”; in line number 142, “Southern” as “the Southern”; in line number, “plant” as “a plant”; in line number 148, “chemical” as “the chemical”; in line number 149, “comparative” as “the comparative”; in line number 149, “composition was” as “composition were”; in line number 153, “the” as “of the”; in line number 154, “high” as “a high”; in line number 157, “higher” as “a higher”; in line number 159, “about hight” as “with the hight”; in line number 160, “lower” as “a lower”; in line number 161, “some of” as “some”; in line number 162, “mixed” as “the mixed”; in line number 165, “strong” as “a strong”; in line number 165, “lower” as “a lower”; in line number 194, “general” as “a general”; in line number 200, “criterium” as “the criterium”; in line number 210, “fraction” as “a fraction”; in line number 234, “The better” as “Better”; in line number 236, “presence” as “the presence”; in line number 239,  “model” as “a model”; in line number 239, “antimicrobial” as “the antimicrobial”; in line number 246, “paper” as “the paper”; in line number 262, “three” as “of three”; in line number 264, “highest” as “the highest”; in line number 273, “anti-staphylococcal” as “the anti-staphylococcal”; in line number 274, “rainy” as “the rainy”; in line number 278, “dry” as “the dry”; in line number 278, “what had crucial” as “which had a crucial”; in line number 278, “influence for” as “influence on”; in line number 287, “Gram-positive” as “the gram-positive”; in line number 288, “for PEs form” as “of PEs from”; in line number 291, “different” as “a different”; in line number 296, “62.5 μg” as “of 62.5 μg”; in line number 296, “antibacterial” as “the antibacterial”; in line number 298, “sample” as “the sample”; in line number 301, “frame” as “the frame”; in line number 301,  “microbiological” as “the microbiological”; in line number 304, “Antibacterial” as “The antibacterial”; in line number 316, “similar” as “a similar”; in line number 316, “what is good” as “which is a good”; in line number 318, “activity” as “the activity”; in line number 322, “, what suggesting” as “, suggesting”; in line number 324, “All among” as “All”; in line number 329, “Bioactivity” as “The bioactivity”; in line number 357, “Presented” as “The presented”; in line number 357, “of evaluation” as “to evaluate”; in line number 358, “treatment” as “the treatment”; in line number 359, “activity” as “the activity”; in line number 360, “ability” as “the ability”; in line number 377, “novel” as “a novel”; in line number 377, “proper” as “the proper”; in line number 380, “0.260” as “of 0.260”; in line number 385, “Marmara” as “the Marmara”; in line number 387, “indicate about” as “indicates”; in line number 392, “crucial” as “a crucial”; in line number 392, “stomach” as “the stomach”; in line number 399, “mortar” as “a mortar”; in line number 417, “previously” as “the previously”; in line number 423, “with 10” as “with a 10”; in line number 443, “final” as “the final”; in line number 454, “10 μL” as “of 10 μL”; in line number 425, “different” as “a different”; in line number 471, “in 4” as “at 4”; in line number 472, “protease” as “a protease”; in line number 472, “Urease” as “The urease”; in line number 474, “extracts was” as “extracts were”; in line number 480, “the ammonia” as “ammonia”; in line number 480, “hydrolysis” as “the hydrolysis”, “calculated” as “was calculated”; in line number 492, “about” as “of about”; in line number 493, “relatively” as “the relatively”; in line number 495, “evaluation of impact” as “an evaluation of the impact”; in line number 499, “classification” as “the classification”; in line number 508, “of isolation” as “to isolate”. The grammar mistakes which are not mentioned hecre are also to be checked and corrected properly.

2. There are some typing mistakes as well, and authors are advised to carefully proof-read the text. For example, in line number 26, the word “papper” may be as “paper,”; in line number 33, “precursors” as “precursor”; in line number 46, “bee-glue” as “bee glue”; in line number 48, “bee-hive” as “bee hive”; in line number 56, “tus” as “check spelling”; in line number 77, “type” as “types”; in line number 77, “infections ,” as “infections,”; in line number 80, “anti-inflamatory” as “anti-inflammatory”; in line number 104, “Generally” as “Generally,”; in line number 107, “reason in” as “reason, in”; in line number 117, “them high” as “them, high”; in line number 118, “but no” as “but not”; in line number 120, “manly” as “mainly”; in line number 126, “non identified” as “non-identified”; in line number 135, “research most” as “research, most”; in line number 139, “occurence” as “occurrence”; in line number 141, “Norther” as “Northern”; in line number 148, “extarcts” as “extracts”; in line number 155, “methylbutenyl” as “methyl butenyl”; in line number 156, “pinobanksin-3-O-cetate” as “pinobanksin-3-O-acetate”; in line number, “cluster” as “cluster,”; in line number 192, “Antimcrobial” as “Antimcrobial”; in line number 214, “antibacteriac” as “antibacterial”; in line number 217, “al PEs” as “all PEs”; in line number 299, “comparsion” as “comparison”; in line number 230, “125 nad” as “125 and”; in line number 230, “Intersingly,” as “Interestingly,”; in line number 230, “dichlromethane” as “dichloromethane”; in line number 234, “comunds” as “compounds”; in line number 234, “extarcts” as “extracts”; in line number 235, “different” as “a different”; in line number 236, “compunds” as “compounds”; in line number 236, “quercitine” as “quercetin”; in line number 236, “naryngenin” as “naringenin”; in line number 238, “Diffrent” as “Different”; in line number 246, “another results” as “another result or other results”; in line number 248, “underline” as “underlined”; in line number 257, “have showed” as “have shown”; in line number 258, “Brazil .” as “Brazil.”; in line number 261, “Another studies” as “Another study or Other studies”; in line number 262, “etanolic” as “ethanolic”; in line number 264, “valus” as “values”; in line number 270, “This findings” as “These findings”; in line number 289, “communicate” as “communication”; in line number 300, “testd” as “tested”; in line number 313, “week” as “weak”; in line number 321, “sub inhibitory” as “sub-inhibitory”; in line number 330, “that our” as “than our”; in line number 343, “and to” as “and”; in line number 360, “shows” as “show”; in line number 379, “inbitory” as “inhibitory”; in line number 380, “similarly” as “similar”; in line number 381, “distincly” as “distinctly”; in line number 382, “example essential” as “example, essential”; in line number 383, “208.3 μg/mL [4517]” as “to 208.3 μg/mL [45 or 45, 17]”; in line number 397, “in in” as “in”; in line number 398, “Kogaly ,” as “Kogaly,”; in line number 402, “hypergrade” as “hyper grade”; in line number 406, “obtained obtained” as “obtained”; in line number 412, “extractionobtained” as “extraction obtained”; in line number 413, “Whattman” as “Whatman”; in line number 427, “dimethylosulfoxide” as “dimethylo sulfoxide”; in line number 466, “microorganisms .” as “microorganisms.”; in line number 492, “bacteriai” as “bacteria”; in line number 506, “poplar” as “popular”. The typos not mentioned here are also to be checked and corrected properly.

3. Check the abbreviations throughout the manuscript and introduce the abbreviation when the full word appears the first time in the abstract and the remaining for the text and then use only the abbreviation (For example, MIC, MBC, uHPLC, propolis extracts (PE), etc.,). Make a word abbreviated in the article that is repeated at least three times in the text, not all words  to be abbreviated. The authors may avoid the usage of abbreviations in keywords.

4. The full form of the species should be given when the first time appears in both the abstract and in the remaining part of the manuscript and it should be followed by only the first letter of the genus (For example, Staphylococcus aureus when the first time appear and followed by S. aureus). It should be checked for all other species mentioned in the manuscript.

5. In the conclusion seems to be in general. All conclusions must be convincing statements on what was found to be novel, impact based on the strong support of the data/results/discussion. And also, it is highly recommended to include limitation of the study.

6. The references are not arranged properly in a uniform format and it should be properly checked and corrected. For example, the authors have given full name of the journals and also short form.

Author Response

Reviewer 4

  1. The English need improvement since there are some grammatical and syntax errors in the manuscript. 

The grammar mistakes which are not mentioned hecre are also to be checked and corrected properly.

A: Thank you very much, for your thorough and detailed review of our manuscript. It helped us a lot in putting it in order

  1. There are some typing mistakes as well, and authors are advised to carefully proof-read the text. For. The typos not mentioned here are also to be checked and corrected properly.

A: As was mentioned above.Thank you for all your suggestions.

  1. Check the abbreviations throughout the manuscript and introduce the abbreviation when the full word appears the first time in the abstract and the remaining for the text and then use only the abbreviation (For example, MIC, MBC, uHPLC, propolis extracts (PE), etc.,). Make a word abbreviated in the article that is repeated at least three times in the text, not all words  to be abbreviated. The authors may avoid the usage of abbreviations in keywords.

A: Thank You for the suggestion. It was corrected.

  1. The full form of the species should be given when the first time appears in both the abstract and in the remaining part of the manuscript and it should be followed by only the first letter of the genus (For example, Staphylococcus aureus when the first time appear and followed by S. aureus). It should be checked for all other species mentioned in the manuscript.

A: Thank You for the suggestion. It was corrected.

  1. In the conclusion seems to be in general. All conclusions must be convincing statements on what was found to be novel, impact based on the strong support of the data/results/discussion. And also, it is highly recommended to include limitations of the study.

 A: Thank You for the suggestion. The conclusion section was corrected and improved.

  1. The references are not arranged properly in a uniform format and it should be properly checked and corrected. For example, the authors have given full name of the journals and also short form.

A: Thank You for the suggestion. It was corrected.

Round 2

Reviewer 1 Report

The authors should pay more attention to technical elements. The tables are not listed appropriately. The content of the table should be rechecked.

Author Response

The authors should pay more attention to technical elements. The tables are not listed appropriately. The content of the table should be rechecked.

A: Thank you for your suggestion. The content of the table has been rechecked. The tables are listed appropriately now.

Reviewer 4 Report

1. There are some grammatical, alignment and typographical errors are noted in the manuscript and it should be thoroughly checked and corrected throughout the manuscript.  For example, in line number 30, the words “used to” may be as “used for”; all over the manuscript “concetration” as “concentration”; in line number 36, “precursors” as “precursor”; in line number 37,  “as main” as “as the main”; in line number 43, “summary” as “summary,”; in line number 49, “bee-glue” as “bee glue”; in line number 51, “bee-hive” as “bee hive”; in line number 59, “profile tus” as “check spelling for tus”; in line number 65, “the Central” as “Central”; in line number 78, “the cosmetology” as “cosmetology”; in line number 81, “type” as “types”; in line number 84, “anti-inflamatory” as “anti-inflammatory”; in line number 85, “agents” as “agent”; in line number 89, “in chemical” as “in the chemical”; in line number 95, “same scopus” as “the same Scopus”; in line number 96, “these investigation” as “this investigation”; in line number 97, “rest” as “the rest”; in line number 98, “, further” as “, and further”; in line number 105, “Accorging” as “According”; in line number 106, “potential” as “the potential”; in line number 106, “factor” as “factors”; in line number 108, “evaluation” as “the evaluation”; in line number 109, “urease” as “the urease”; in line number 115, “was” as “were”; in line number 121, “components” as “the components”; in line number 122, “components” as “the components”; in line number 123, “allow” as “allows”; in line number 123, “above” as “the above”; in line number 125, “For this reason in the current paper” as “For this reason, the current paper,”; in line number 131, “components” as “components,”; in line number 133, “Usually” as “Usually,”; in line number 136, “they was” as “they were”; in line number 137, “components was” as “the components were”; in line number 143, “them high” as “them, high”; in line number 151, “components” as “the components”; in line number 152, “known such” as “known”; in line number 159,  “presence” as “the presence”; in line number 160, “specific” as “a specific”; in line number 162, “presented” as “present”; in line number 162, “China it” as “China, it”; in line number 163, “garden” as “gardens”; in line number 164, “natural environment” as “the natural environments”; in line number 165, “than” as “more than”; in line number 167, “current” as “the current”; in line number 173, “the Northern” as “Northern”; in line number 174, “sample” as “the sample”; in line number 179, “cultivars is” as “cultivars are”; in line number 180, “from easy” as “for the easy”; in line number 180, “In a result in same Almaty” as “As a result in the same Almaty,”; in line number 185, “Populus genus” as “the Populus genus,”; in line number 188, “trace” as “a trace”; in line number, “same” as “the  same”; in line number 192, “Possibility” as “The possibility”; in line number 193,  “expeciallly” as “expecially”; in line number 195, “what may suggested” as “which may suggest”; in line number 197, “a different” as “different”; in line number 198, “honeybee” as “honeybees”; in line number 199, “environment” as “the environment”; in line number 199, “presence of uncommon component” as “the presence of an uncommon components”; in line number 222, “Results of comparative” as “The results of the comparative”; in line number 223, “Investigation” as “An investigation”; in line number 223, “uHPLC-DAD” as “the uHPLC-DAD”; in line number 224, “the samples in” as “of the samples into”; in line number 228, “high” as “a high”; in line number 231, “higher” as “a higher”; in line number 232, “lower of” as “lower”; in line number 233, “was possible” as “were possible”; in line number 235, “about strong” as “wioth a strong”; in line number 236, “lower” as “a lower”; in line number 237, “some of” as “some”; in line number 238, “mixed” as “a mixed”; in line number 241, “lower” as “a lower”; in line number 243, “summary” as “summary,”; in line number 243, “dendrogram reflected presence” as “the dendrogram reflected the presence”; in line number 244, “descrbied” as “described”; in line number 244, “presence” as “the presence”; in line number 244, “sameples” as “samples”; in line number 246, “low” as “a low”; in line number 247, “suspecpet” as “suspect”; in line number 247, “Populus” as “the Populus”; in line number 247, “main” as “the main”; in line number 251, “general” as “a general”; in line number 252, “study we evaluated activity” as “study, we evaluated the activity”; in line number 257, “criterium” as “the criterium”; in line number 261, “Distinguish” as “Distinguished”; in line number 261, “rest” as “the rest”; in line number 261, “own” as “its own”; in line number 262, “what suggested presence” as “which suggested the presence”; in line number 272, “fraction” as “a fraction”; in line number 279, “al 70EEPs” as “all 70EEPs”; in line number 282, “the gastrointestinal” as “gastrointestinal”; in line number 434, “ability” as “the ability”; in line number 436, “suggested, that same” as “suggested that the same”; in line number 437 and 439, “direct” as “directly”; in line number 439, “same” as “the same”; in line number 441, “complex” as “a complex”; in line number 442, “reason some” as “reason, some”; in line number 460, “Inhibition” as “The inhibition”; in line number 494, “the previously” as “previously”; in line number 505, “Scan” as “The scan”; in line number 506, “dimethylo” as “dimethyl”; in line number 545, “different” as “a different”; in line number 546, “microorganisms .” as “microorganisms.”; in line number 561, “the ammonia” as “ammonia”; in line number 563, “Activity” as “The activity”; in line number 570, “Analysis” as “The analysis”; in line number 572, “relatively” as “relative”; in line number 573, “about” as “of”; in line number 573, “matrix” as “the matrix”; in line number 581, “poplar” as “popular”; in line number 584, “was ambiguous” as “were ambiguous”; in line number 585, “presence additional” as “the presence of additional”; in line number 585, “precrusors” as “precursors”.

2. The authors have carried out the suggested correction properly. The full form of the species should be given when the first time appears in both the abstract and in the remaining part of the manuscript and it should be followed by only the first letter of the genus (For example, in line number 258 the full form is given for “Staphylococcus aureus” and also the same has given in line number 514 also and the same may be replaced by S. aureus). It should be checked for all other species mentioned in the manuscript.

Author Response

  1. There are some grammatical, alignment and typographical errors are noted in the manuscript and it should be thoroughly checked and corrected throughout the manuscript. 

A: Thank you for your suggestion. The content of the table has been rechecked. The tables are listed  appropriately now.

  1. The authors have carried out the suggested correction properly. The full form of the species should be given when the first time appears in both the abstract and in the remaining part of the manuscript and it should be followed by only the first letter of the genus (For example, in line number 258 the full form is given for “Staphylococcus aureus” and also the same has given in line number 514 alsoand the same may be replaced byS. aureus). It should be checked for all other species mentioned in the manuscript.

A: Thank you for your suggestion. Names of all species have been checked and corrected.

We would like to thank the reviewer for her/his insightful and very helpful review. We would very much like to cooperate scientifically with such a reliable scholar in the future.
